# Convexity-Preserving Rational Cubic Zipper Fractal Interpolation Curves and Surfaces

**Vijay \***  **and Arya Kumar Bedabrata Chand**

Department of Mathematics, Indian Institute of Technology Madras, Chennai 600036, India
* Correspondence: vijaysiwach975@gmail.com

**Abstract:** A class of zipper fractal functions is more versatile than corresponding classes of traditional and fractal interpolants due to a binary vector called a signature. A zipper fractal function constructed through a zipper iterated function system (IFS) allows one to use negative and positive horizontal scalings. In contrast, a fractal function constructed with an IFS uses positive horizontal scalings only. This article introduces some novel classes of continuously differentiable convexity-preserving zipper fractal interpolation curves and surfaces. First, we construct zipper fractal interpolation curves for the given univariate Hermite interpolation data. Then, we generate zipper fractal interpolation surfaces over a rectangular grid without using any additional knots. These surface interpolants converge uniformly to a continuously differentiable bivariate data-generating function. For a given Hermite bivariate dataset and a fixed choice of scaling and shape parameters, one can obtain a wide variety of zipper fractal surfaces by varying signature vectors in both the $x$ direction and $y$ direction. Some numerical illustrations are given to verify the theoretical convexity results.

**Keywords:** iterated function system; fractals; convexity; rational cubic spline; blending functions; convergence; zipper; zipper fractal surfaces

## 1. Introduction

The technique of constructing fractal interpolation functions (FIFs) initiated by Barnsley [1] can produce nonsmooth or smooth interpolants [2], where the graph of an FIF is an attractor of a suitable IFS. This technique involves a free parameter to control the variation of ordinates so that it provides flexibility to generate a range of interpolants from smooth to nowhere differentiable on a compact domain. A class of spline interpolants can be generalized using the class of smooth fractal interpolants. Implementing the idea of spline FIFs given by Barnsley and Harrington [2], cubic spline FIFs with general boundary conditions were introduced by Chand and Kapoor [3]. Since the classical splines are used in the problem of shape-preserving interpolation, it is natural to think that their fractal generalization can also do the same. Chand et al. in [4] introduced rational cubic FIFs, which can preserve the fundamental features of univariate Hermite data.

In the literature, univariate FIFs are studied more than fractal interpolation surfaces (FISs). It may be the case that the graph of a linear FIS is not continuous [5], whereas the graph of a linear fractal interpolation function (one-dimensional) is always continuous [1]. Massopust [6] proposed the construction of FISs on triangular domains. He used coplanar interpolation points on the boundary of the domain for constructing the FISs. Geronimo and Hardin in [7] presented the constructions of FISs on flexible domains. Zhao [8] generalized that construction using barycentric co-ordinates. In [9,10], the researchers generated FISs on rectangular grids for collinear interpolation points on the boundary. The construction of an FIS given in [10] was improved in [9]. A new construction for FISs for every set of data and a generalization to higher dimensions is given in [11]. FISs with recurrent IFSs were constructed in [12,13]. Navascués et al., in [14], analyzed the spanning properties of the fractal functions on the rectangle. Chand et al., in [15], investigated FISs that lie above

a plane. Fractal surfaces have been found to be effective in scientific applications, such as metallurgy, geology, computer graphics, physics, brain electrical activity, and image processing, e.g., [16–19].

The idea of a zipper, which generalizes the concept of an IFS using a binary vector named signature, was given by Aseev et al. in [20]. Tetenov and collaborators studied many interesting topological and structural properties of zippers related to dendrites and self-similar continua in [21–25]. Chand et al. in [26] generated affine zipper FIFs and examined their range-restricted properties. Recently, the approximation features of smooth zipper fractal functions have been studied in [27]. In this article, we construct one-dimensional and two-dimensional continuously differentiable zipper fractal interpolants. To construct a class of continuously differentiable zipper fractal surface interpolants, we use the technique given in [28], where the horizontal scalings in the $x$ and $y$ directions can be negative. This technique of constructing surface interpolants using the network of univariate boundary curves and blending functions is beneficial in the design environment. Since the generated surface inherits all the properties of the network of boundary curves, see [29], we generate convexity-preserving surface interpolants using some restriction on the zipper IFS parameters. Our scheme can produce a wide variety of surface interpolants due to the presence of a signature, scaling factors, and shape parameters. It is more convenient to model smooth convex bivariate interpolants in shape abstraction and modeling, computer-aided design, biomedical instruments, object recognition, computer graphics, reverse engineering material science, and metallurgy to capture the irregularities associated with the partial derivatives of the surface interpolants.

## 2. Basics of Fractal Interpolation

We shall begin with the basics of fractal interpolation, and fractal perturbation of a function through $\alpha$-fractal functions. The details can be found in the references [1,2,30].

Consider a finite interpolation dataset $\{(x_i, y_i) \in \mathbb{R}^2 : i \in \mathbb{N}_N := \{1, 2, \dots, N\}\}$, where $x_1 < x_2 < \dots < x_N$ and $N > 2$. Let $I := [x_1, x_N]$ and $I_i := [x_i, x_{i+1}]$ for $i \in \mathbb{N}_{N-1}$. Let $\mathcal{L}_i : I \to I_i$ be homeomorphisms such that for all $x, x^* \in I$ and for some $0 \le l_i < 1$,

$$|\mathcal{L}_i(x) - \mathcal{L}_i(x^*)| \le l_i |x - x^*|; \quad \mathcal{L}_i(x_1) = x_i, \ \mathcal{L}_i(x_N) = x_{i+1}, \ i \in \mathbb{N}_{N-1}. \quad (1)$$

Define $N - 1$ continuous maps $\mathcal{F}_i : I \times \mathbb{R} \to \mathbb{R}$ such that

$$|\mathcal{F}_i(x, y) - \mathcal{F}_i(x, y^*)| \le |\alpha_i| |y - y^*|; \quad \mathcal{F}_i(x_1, y_1) = y_i, \ \mathcal{F}_i(x_N, y_N) = y_{i+1}, \ i \in \mathbb{N}_{N-1}, \quad (2)$$

for all $x \in I$, $y, y^* \in \mathbb{R}$, and for some $\alpha_i \in (-1, 1)$. Now construct the elementary functions

$$\mathcal{W}_i : I \times \mathbb{R} \to I_i \times \mathbb{R} \subseteq I \times \mathbb{R}, \quad \mathcal{W}_i(x, y) = \big(\mathcal{L}_i(x), \mathcal{F}_i(x, y)\big), \quad i \in \mathbb{N}_{N-1}.$$

**Theorem 1** ([1]). *The iterated function system (IFS) $\{I \times \mathbb{R}; \mathcal{W}_i, i \in \mathbb{N}_{N-1}\}$ has a unique attractor $\mathcal{G}$, which is the graph of a continuous function $g^* : I \to \mathbb{R}$ satisfying $g^*(x_i) = y_i$ for all i. Furthermore, if $\widehat{C}(I) := \{g : I \to \mathbb{R}, g \text{ is continuous on } I, g(x_1) = y_1, g(x_N) = y_N\}$ is endowed with the uniform metric, and the Read–Bajraktarević (RB) operator $T^\alpha : \widehat{C}(I) \to \widehat{C}(I)$ is defined by $T^\alpha g(x) = \mathcal{F}_i\big(\mathcal{L}_i^{-1}(x), g \circ \mathcal{L}_i^{-1}(x)\big), x \in I_i, i \in \mathbb{N}_{N-1}$, then $g^*$ is the unique fixed point of $T^\alpha$.*

The function $g^*$ in the above theorem is called an FIF corresponding to the interpolation dataset $\{(x_i, y_i) \in I \times \mathbb{R} : i = 1, 2, \dots, N\}$. The popular fractal interpolation in theory and applications is taken from the following IFS:

$$\{I \times \mathbb{R}; \mathcal{W}_i = \big(\mathcal{L}_i(x), \mathcal{F}_i(x, y)\big), i \in \mathbb{N}_{N-1}\}; \quad \mathcal{L}_i(x) = a_i x + b_i, \ \mathcal{F}_i(x, y) = \alpha_i y + q_i(x). \quad (3)$$

Here, $\alpha_i$ is called the vertical scaling factor corresponding to the map $\mathcal{W}_i$. The corresponding fractal function $g^*$ satisfies

$$g^*(x) = \alpha_i\big(\mathcal{L}_i^{-1}(x)\big) g^*\big(\mathcal{L}_i^{-1}(x)\big) + q_i\big(\mathcal{L}_i^{-1}(x)\big), \quad x \in I_i, \quad i \in \mathbb{N}_{N-1}.$$

Let $C^k(I)$ be a collection of all $k$-times continuously differentiable real-valued functions on $I$. The existence of differentiable fractal functions or spline FIFs is given in the following:

**Theorem 2** ([2]). *Let $\{(x_i, y_i), i = 1, 2, \ldots, N\}$ be a given dataset, where $x_1 < \ldots < x_N$. Suppose that $\mathcal{L}_i(x) = a_i x + b_i$, $\mathcal{F}_i(x, y) = \alpha_i y + q_i(x)$, where $q_i$ is continuous for $i \in \mathbb{N}_{N-1}$. Suppose for some integer $p \geq 0$, $|\alpha_i| < a_i^p$, $i \in \mathbb{N}_{N-1}$. Let $\mathcal{F}_i^k(x, y) = \frac{\alpha_i y + q_i^{(k)}(x)}{a_i^k}$, $q_i^{(k)}(x)$ represents the $k^{th}$ derivative of $q_i(x)$,*

$$y_1^k = \frac{q_1^{(k)}(x_1)}{a_1^k - \alpha_1}, \quad y_N^k = \frac{q_{N-1}^{(k)}(x_N)}{a_{N-1}^k - \alpha_{N-1}}, \quad k = 1, 2, \ldots, p.$$

*If $\mathcal{F}_i^k(x_N, y_N^k) = \mathcal{F}_{i+1}^k(x_1, y_1^k)$, $i \in \mathbb{N}_{N-2}$, $k = 1, 2, \ldots, p$,*

*then the IFS $\{I \times \mathbb{R}; \mathcal{W}_i(x, y) = (\mathcal{L}_i(x), \mathcal{F}_i(x, y)), i \in \mathbb{N}_{N-1}\}$ determines a spline FIF $g^* \in C^p[x_1, x_N]$, and $g^{*(k)}$ is the FIF determined by $\{I \times \mathbb{R}; \mathcal{W}_i(x, y) = (\mathcal{L}_i(x), \mathcal{F}_i^k(x, y)), i \in \mathbb{N}_{N-1}\}$.*

If $q_i(x)$'s are taken as rational functions, one can obtain rational fractal splines, see for instance [4,31].

The concept of FIFs can be used to associate a family of continuous fractal functions with a prescribed function $f \in C(I)$ (see [30]). For this procedure, consider a partition $\pi_\Delta := \{x_1, x_2, \ldots, x_N\}$ of $I$ with increasing abscissae. Suppose that $b \in C(I)$ satisfies $b(x_1) = f(x_1)$ and $b(x_N) = f(x_N)$. For $i \in \mathbb{N}_{N-1}$, we construct $q_i(x) := f(\mathcal{L}_i(x)) - \alpha_i b(x)$, and then the corresponding RB-operator provides a fixed point denoted by $f_{\Delta,b}^\alpha = f^\alpha$, where $\alpha = (\alpha_1, \alpha_2, \ldots, \alpha_{N-1}) \in (-1, 1)^{N-1}$. The associate fractal function $f^\alpha$ is called the $\alpha$-fractal function corresponding to $f$ and it satisfies the self-referential equation:

$$f^\alpha(x) = f(x) + \alpha_i(f^\alpha - b)(\mathcal{L}_i^{-1}(x)), \quad x \in I_i, \ i \in \mathbb{N}_{N-1}.$$

This model is used to construct a fractal perturbation of any standard function in numerical analysis and approximation theory.

For a $C^p(I)$-fractal perturbation function $f^\alpha$ from a germ $f$ in the same space, we proceed as follows: Let $f \in C^p(I)$, and $|\alpha_i| < a_i^p, i \in \mathbb{N}_{N-1}$. In order to construct $f^\alpha \in C^p(I)$ corresponding to $f$, it is sufficient to find the conditions on the base function $b$ according to Theorem 2.

**Theorem 3** ([32]). *Let $f \in C^p(I)$, and $x_1 < x_2 < \ldots < x_N$ be an arbitrary partition of $I$. Suppose $|\alpha_i| < a_i^p$ for all $i \in \mathbb{N}_{N-1}$. Further, assume that the base function $b$ obeys*

$$b^{(k)}(x_1) = f^{(k)}(x_1), \ b^{(k)}(x_N) = f^{(k)}(x_N) \text{ for } k = 0, 1, \ldots, p.$$

*Then, the corresponding fractal function $f^\alpha$ is $p$-times differentiable, and $(f^\alpha)^{(k)}(x_i) = f^{(k)}(x_i)$ for $i \in \mathbb{N}_N; k = 0, 1, \ldots, p$.*

In the next section, we will take $f$ as a zipper rational cubic spline, and then we will perturb it to construct the desired zipper fractal rational cubic spline as $f^\alpha$. Since it is difficult for a zipper fractal spline constructed using a single base function to preserve the shape of the data, we will use a family of base functions satisfying the conditions given in Theorem 3 to perturb a zipper rational cubic spline.

## 3. Construction of Univariate Hermite Zipper RCS and Its Fractal

A classical spline can be written as a function from the entire interval $[x_1, x_N]$ to the subinterval $[x_{i-1}, x_i]$, and the contraction factor is always positive. If the interval is also allowed to map in the reverse direction to any subinterval, i.e., $x_1$ maps to $x_i$ and $x_N$ maps to $x_{i-1}$, then the horizontal contraction factor will be negative. A classical spline can be

extended in this manner using a positive contraction or a negative contraction in each subinterval. Thus, we need a binary signature to fulfill such options, and hence the zipper. We construct a family of novel continuously differentiable zipper rational cubic splines (RCSs) using a binary vector signature in Section 3.1 in the first step. The rational cubic spline is a member of this family. In Section 3.2, we generate a new class of continuously differentiable zipper fractal interpolants by fractalizing each member of the family of zipper RCSs using a suitable family of base functions.

### 3.1. Construction of Hermite Zipper RCS

Consider a finite set of Hermite interpolation data $\{(x_i, y_i, d_i) \in I \times \mathbb{R}^2 : i \in \mathbb{N}_N\}$, where $y_i$ and $d_i$ are the function value and the first derivative value at the knot $x_i$, respectively, $I = [x_1, x_N]$ with $x_1 < x_2 < \ldots < x_N$. For all $i \in \mathbb{N}_N$, assume that $y_i \in [\varrho_1, \varrho_2]$, for some $\varrho_1, \varrho_2 \in \mathbb{R}$, and $d_i$ are either given or calculated from data by some appropriate methods given in [4]. For a fixed signature $\epsilon := (\epsilon_1, \epsilon_2, \ldots, \epsilon_{N-1}) \in \{0, 1\}^{N-1}$, let $\mathcal{L}_i : I \to I_i = [x_i, x_{i+1}], i = 1, 2, \ldots, N-1$, be contractive homeomorphisms such that

$$\begin{aligned} \mathcal{L}_i(x_1) = x_{i+\epsilon_i}, \quad \mathcal{L}_i(x_N) = x_{i+1-\epsilon_i}, \quad \mathcal{L}_i(x) = a_i x + b_i, \\ |\mathcal{L}_i(x) - \mathcal{L}_i(x^*)| \leq l_i |x - x^*|, \quad \forall x, x^* \in I, \end{aligned} \tag{4}$$

for some $0 \leq l_i < 1$.

Let $h_i^* := x_{i+1-\epsilon_i} - x_{i+\epsilon_i}$, $h_i := x_{i+1} - x_i$, $\Delta_i := \frac{y_{i+1} - y_i}{x_{i+1} - x_i}$, $i \in \mathbb{N}_{N-1}$, and $|I| := x_N - x_1$. For $0 \leq \theta := \frac{x - x_1}{x_N - x_1} \leq 1$, $\sigma_i > 0$, and $\eta_i > 0$, let $Q_i(\theta) = \sigma_i(1 - \theta)^2 + 2\sigma_i \eta_i (1 - \theta)\theta + \eta_i \theta^2$. Now, consider a rational function with a cubic numerator and a quadratic denominator of the form:

$$R_\epsilon(\mathcal{L}_i(x)) = \frac{R_i(\theta)}{Q_i(\theta)}, \ i \in \mathbb{N}_{N-1}, \ x \in I, \tag{5}$$

where

$$\begin{aligned} R_i(\theta) &= \sum_{k=0}^{3} A_{ik}(1-\theta)^{3-k}\theta^k, \\ A_{i0} &= \sigma_i y_{i+\epsilon_i}, \quad A_{i1} = (2\sigma_i \eta_i + \sigma_i)y_{i+\epsilon_i} + \sigma_i h_i^* d_{i+\epsilon_i}, \\ A_{i2} &= (2\sigma_i \eta_i + \eta_i)y_{i+1-\epsilon_i} - \eta_i h_i^* d_{i+1-\epsilon_i}, \quad A_{i3} = \eta_i y_{i+1-\epsilon_i}. \end{aligned}$$

One can easily obtain the following results:

**Theorem 4.** *Let $\{(x_i, y_i, d_i) \in I \times \mathbb{R}^2 : i \in \mathbb{N}_N\}$ be a Hermite interpolation dataset. For the fixed signature $\epsilon := (\epsilon_1, \epsilon_2, \ldots, \epsilon_{N-1}) \in \{0, 1\}^{N-1}$, the rational function $R_\epsilon$ defined in (5) has the following properties:*

*(i)    $R_\epsilon$ is a member of the space $C^1(I)$;*

*(ii)   $R_\epsilon$ interpolates the given Hermite data, i.e., $R_\epsilon(x_i) = y_i$ and $R'_\epsilon(x_i) = d_i$ for all $i \in \mathbb{N}_N$.*

We call this rational function $R_\epsilon$ as zipper RCS for the given Hermite data. These $\sigma_i$ and $\eta_i$, $i \in \mathbb{N}_{N-1}$, which we used to construct Hermite zipper RCS, are called shape parameters, and we call $\sigma = (\sigma_1, \sigma_2, \ldots, \sigma_{N-1})$ and $\eta = (\eta_1, \eta_2, \ldots, \eta_{N-1})$ shape parameter vectors. For the given Hermite interpolation data $\{(x_i, y_i, d_i) \in I \times \mathbb{R}^2 : i \in \mathbb{N}_N\}$, if we take $\sigma_i$ and $\eta_i$ such that $\sigma_i \neq \eta_i$ for all $i \in \mathbb{N}_{N-1}$, then using different values of signature (we have $2^{N-1}$ number of choices to choose a signature), one can construct $2^{N-1}$ distinct Hermite zipper RCSs. The proposed class of these Hermite zipper RCSs takes the classical rational cubic spline (when $\epsilon \equiv 0$) defined in [33] and the piecewise zipper cubic polynomial (when $\sigma_i = \eta_i = 1, \forall i$) as particular cases.

*3.2. Construction of a $C^1$-RCS ZFIF*

In this subsection, we generate $\alpha$-fractal functions corresponding to $R_\epsilon$ using Theorem 3.

For a fixed signature $\epsilon := (\epsilon_1, \epsilon_2, \ldots, \epsilon_{N-1}) \in \{0,1\}^{N-1}$, let $R_\epsilon$ defined in (5) be a zipper RCS for the given Hermite interpolation data $\{(x_i, y_i, d_i) \in I \times \mathbb{R}^2 : i \in \mathbb{N}_N\}$. Consider a set of $C^1$-functions $\{\mathcal{B}_i : i \in \mathbb{N}_{N-1}\}$ such that

$$\mathcal{B}_i(x) = \frac{R_i^*(\theta)}{Q_i(\theta)}, \ i \in \mathbb{N}_{N-1}, \tag{6}$$

where

$$R_i^*(\theta) = \sum_{k=0}^{3} A_{ik}^*(1-\theta)^{3-k}\theta^k,$$

$$A_{i0}^* = \sigma_i y_1, \ \ A_{i1}^* = (2\sigma_i \eta_i + \sigma_i)y_1 + \sigma_i|I|d_1,$$

$$A_{i2}^* = (2\sigma_i \eta_i + \eta_i)y_N - \eta_i|I|d_N, \ \ A_{i3}^* = \eta_i y_N.$$

Easily, we can obtain $\mathcal{B}_i(x_k) = y_k$ and $\mathcal{B}_i'(x_k) = d_k$ for $k = 1, N$. We can take this family of base functions $\mathcal{B}_i$, $i \in \mathbb{N}_{N-1}$ to construct $N-1$ continuous functions $\mathcal{F}_i : \mathcal{K} := I \times [\varrho_1, \varrho_2] \to \mathbb{R}$ such that

$$\mathcal{F}_i(x,y) = \alpha_i y + \big(R_\epsilon(\mathcal{L}_i(x)) - \alpha_i \mathcal{B}_i(x)\big),$$

where $-1 < \alpha_i < 1$. For each $i \in \mathbb{N}_{N-1}$, we have

$$\mathcal{F}_i(x_1, y_1) = y_{i+\epsilon_i}, \ \ \mathcal{F}_i(x_N, y_N) = y_{i+1-\epsilon_i},$$
$$|\mathcal{F}_i(x,y) - \mathcal{F}_i(x, y^*)| \le |\alpha_i||y - y^*|, \ \ \forall x \in I, y, y^* \in [\varrho_1, \varrho_2]. \tag{7}$$

Define mappings $\mathcal{W}_i : \mathcal{K} \to I_i \times \mathbb{R}$, $i = 1, 2, \ldots, N-1$ by

$$\mathcal{W}_i(x,y) = \big(\mathcal{L}_i(x), \mathcal{F}_i(x,y)\big), \ \ \forall (x,y) \in \mathcal{K}.$$

**Definition 1.** *Let $\mathcal{X}$ be a complete metric space. For given vertices $v_1, v_2, \ldots, v_N \in \mathcal{X}$ and signature $\epsilon = (\epsilon_1, \epsilon_2, \ldots, \epsilon_{N-1}) \in \{0,1\}^{N-1}$, a zipper IFS is a collection of $\mathcal{X}$ with some contraction maps on $\mathcal{X}$ to itself, which is denoted by $\mathcal{I} := \{\mathcal{X}; \mathcal{Z}_i : i \in \mathbb{N}_{N-1}\}$, where $\mathcal{Z}_i(v_1) = v_{i+\epsilon_i}$ and $\mathcal{Z}_i(v_N) = v_{i+1-\epsilon_i}$ for all $i \in \mathbb{N}_{N-1}$. The compact set $\mathcal{G} \subset X$, which satisfies*

$$\mathcal{G} = \bigcup_{i=1}^{N-1} \mathcal{Z}_i(\mathcal{G}),$$

*is called the attractor or fractal corresponding to the zipper IFS $\mathcal{I}$.*

Since $\mathcal{K} := I \times [\varrho_1, \varrho_2]$ is a complete metric space with the usual metric and each $\mathcal{W}_i$ is a contraction mapping such that $\mathcal{W}_i(x_1, y_1) = (x_{i+\epsilon_i}, y_{i+\epsilon_i})$ and $\mathcal{W}_i(x_N, y_N) = (x_{i+1-\epsilon_i}, y_{i+1-\epsilon_i})$, then $\mathcal{I} = \{\mathcal{K}; \mathcal{W}_i : i \in \mathbb{N}_{N-1}\}$ becomes a zipper IFS with vertices $(x_1, y_1), (x_2, y_2), \ldots, (x_N, y_N)$ and signature $\epsilon = (\epsilon_1, \epsilon_2, \ldots, \epsilon_{N-1})$. Since $R_\epsilon$ and $\mathcal{B}_i$ for each $i \in \mathbb{N}_{N-1}$ belong to $C^1(I)$ and agree on the endpoints of subinterval $I$, if we choose $|\alpha_i| < |a_i|$, then the zipper IFS $\mathcal{I}$ determines a unique attractor $\mathcal{G}$, which is a graph of a continuously differentiable function , say $R_\epsilon^\alpha$, on $I$. $R_\epsilon^\alpha$ interpolates the given Hermite data, i.e., $R_\epsilon^\alpha(x_i) = y_i$ and $R_\epsilon^{\alpha(1)}(x_i) = d_i$ for all $i = 1, 2, \ldots, N$, see [27]. We name this $R_\epsilon^\alpha$ as a rational cubic spline zipper fractal interpolation function (RCS ZFIF). It can also be called $\alpha$-fractal function corresponding to $R_\epsilon$. One can observe that $R_\epsilon^\alpha$ is a unique fixed point of the RB operator $T_\epsilon^\alpha : \widehat{C^1}(I) \to \widehat{C^1}(I)$ defined as

$$T_\epsilon^\alpha g(\mathcal{L}_i(x)) = R_\epsilon(\mathcal{L}_i(x)) + \alpha_i\big(g(x) - \mathcal{B}_i(x)\big), \ x \in I, i = 1, 2, \ldots, N-1, \tag{8}$$

which is a contraction map on $\widehat{C^1}(I)$, where $\widehat{C^1}(I) := \{g \in C^1(I) : g(x_1) = y_1, g(x_N) = y_N, g'(x_1) = d_1, \text{ and } g'(x_N) = d_N\}$ is a closed subspace of a complete metric space $C^1(I)$ with respect to $C^1$-norm. Therefore, $R_\epsilon^\alpha \in C^1(I)$ and satisfies

$$R_\epsilon^\alpha(\mathcal{L}_i(x)) = R_\epsilon(\mathcal{L}_i(x)) + \alpha_i(R_\epsilon^\alpha(x) - \mathcal{B}_i(x)), \ \forall x \in I, \ i = 1, 2, \ldots, N - 1. \tag{9}$$

From (9), one can deduce that $R_\epsilon^\alpha$ interpolates the given Hermite data. The class of RCS ZFIFs generalizes the class of Hermite zipper RCSs defined in (5) and the class of rational cubic spline fractal interpolation functions (RCS FIFs).

**Example 1.** *Consider a univariate Hermite interpolation dataset*

$$\{(-2, 1, 3), (-1, -2, 1), (0, 2, -2), (1, -1, 1), (2, 3, -4)\}.$$

*Using the parameters given in Table 1, we plotted the proposed univariate interpolants in Figure 1a–f. In Figure 1a–c, we generated zipper RCSs using different values of signature with the same values of shape parameters. One can notice that these zipper RCSs are different functions. Since we have $2^{5-1} = 16$ choices to choose a signature for the given data, we can construct 16 different zipper RCSs. Hence, the class of zipper RCSs extends the class of classical RCSs. Taking $\epsilon = (0, 0, 0, 0)$, we constructed the classical RCS in Figure 1c. To show the effects of shape parameters, we constructed Figure 1a,d using different values of $\sigma_1$ only. From their plots, it can be observed that, if we take a large value of shape parameter on some subinterval, then the corresponding plot on that subinterval becomes nearly a straight line. Figure 1e,f are the graphs of the RCS ZFIFs. We have plotted these graphs using different values of $\alpha_3$, and one can notice the changes in subinterval $[0, 1]$.*

**Table 1.** Zipper IFS parameters for Figure 1.

| Figure 1 | $\sigma$ | $\eta$ | $\epsilon$ | $\alpha$ |
|---|---|---|---|---|
| (a) | $(0.1, 0.2, 0.1, 0.2)$ | $(1, 2, 2, 1)$ | $(1, 1, 0, 1)$ | $--$ |
| (b) | $(0.1, 0.2, 0.1, 0.2)$ | $(1, 2, 2, 1)$ | $(0, 1, 0, 1)$ | $--$ |
| (c) | $(0.1, 0.2, 0.1, 0.2)$ | $(1, 2, 2, 1)$ | $(0, 0, 0, 0)$ | $--$ |
| (d) | $(2, 0.2, 0.1, 0.2)$ | $(1, 2, 2, 1)$ | $(1, 1, 0, 1)$ | $--$ |
| (e) | $(2, 0.2, 0.1, 0.2)$ | $(1, 2, 2, 1)$ | $(1, 1, 0, 1)$ | $(0.2, -0.15, 0.1, 0.24)$ |
| (f) | $(2, 0.2, 0.1, 0.2)$ | $(1, 2, 2, 1)$ | $(1, 1, 0, 1)$ | $(0.2, -0.15, -0.2, 0.24)$ |

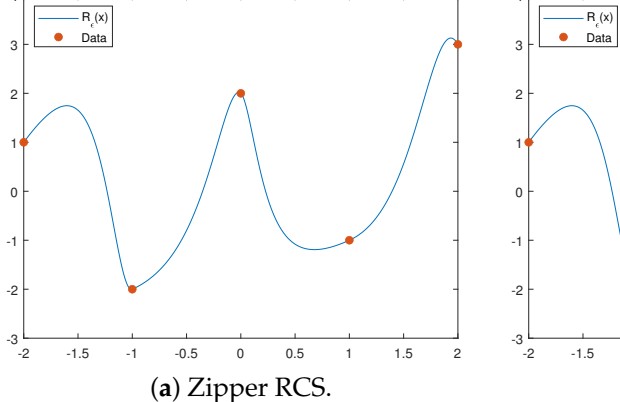

(**a**) Zipper RCS.

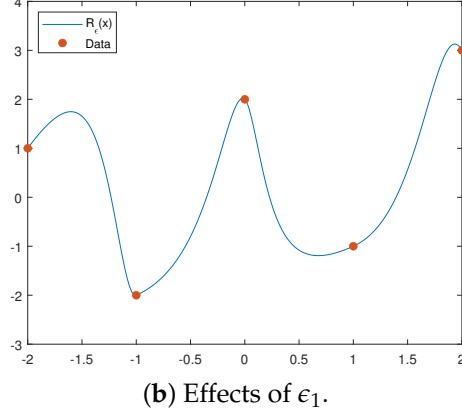

(**b**) Effects of $\epsilon_1$.

**Figure 1.** *Cont.*

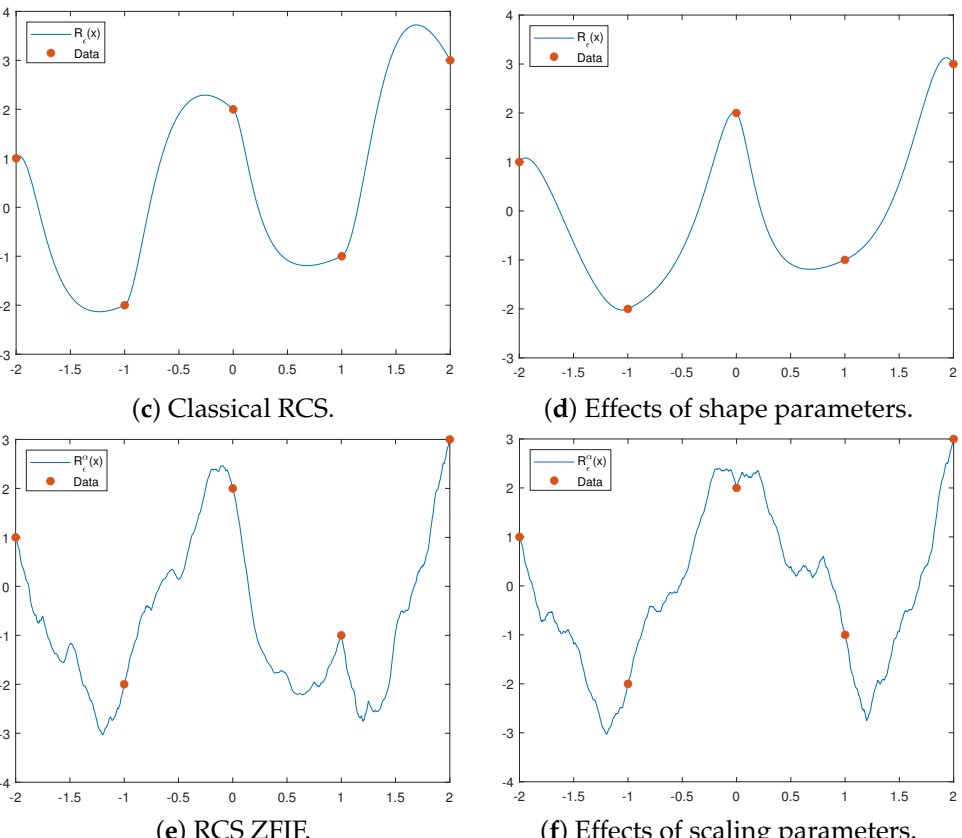

**(c)** Classical RCS.

**(d)** Effects of shape parameters.

**(e)** RCS ZFIF.

**(f)** Effects of scaling parameters.

**Figure 1.** Univariate interpolants.

## 4. Convexity Preserving Zipper RCS and RCS ZFIF

For a convex Hermite dataset $\{(x_i, y_i, d_i) : i = 1, 2, \ldots, N\}$, where $d_1 < \Delta_1 < \ldots < d_i < \Delta_i < d_{i+1} < \Delta_{i+1} < \ldots < \Delta_{N-1} < d_N$, first we will derive sufficient conditions on the shape parameters so that the corresponding zipper rational cubic spline $R_\epsilon$ becomes convex on $I$. Then, we will give sufficient conditions on the zipper IFS parameters so that the proposed RCS ZFIF $R_\epsilon^\alpha$ becomes convex on $I$. Since the first derivative of the proposed RCS ZFIF can be nowhere differentiable on $I$, to prove the convexity of the RCS ZFIF, we will show that its right-handed double derivative or the left-handed double derivative on each point of the interpolating interval exists and is non-negative.

**Proposition 1** ([34])**.** *For a continuous function $g$ on $I = [x_1, x_N]$ and for each $x \in (x_1, x_N)$, if one of the one-sided derivatives $g^{(1)}(x^+)$ or $g^{(1)}(x^-)$ exists, and is non-negative (can be $+\infty$), then the continuous function $g$ is monotonically increasing on $I$.*

We use the above proposition for the first derivative of the proposed interpolant to preserve the convexity feature of the data.

**Theorem 5.** *Let $\{(x_i, y_i, d_i) : i = 1, 2, \ldots, N\}$ be a convex Hermite dataset, where $d_1 < \Delta_1 < \ldots < d_i < \Delta_i < d_{i+1} < \Delta_{i+1} < \ldots < \Delta_{N-1} < d_N$. For a fixed signature $\epsilon \in \{0, 1\}^{N-1}$, if the shape parameters are chosen such that*

$$\sigma_i > \max\left\{0, \frac{d_{i+1-\epsilon_i} - \Delta_i}{2(\Delta_i - d_{i+\epsilon_i})}\right\}, \quad \eta_i > \max\left\{0, \frac{\Delta_i - d_{i+\epsilon_i}}{2(d_{i+1-\epsilon_i} - \Delta_i)}\right\}, \quad i = 1, 2, \ldots, N-1,$$

*then the corresponding zipper RCS will be convex on $I$.*

**Proof.** From the construction (5), $R_\epsilon^{(2)}$ may be discontinuous at the internal grids. In addition, $R_\epsilon^{(2)}(x_i^-)$ and $R_\epsilon^{(2)}(x_i^+)$ exist but may be $R_\epsilon^{(2)}(x_i^-) \neq R_\epsilon^{(2)}(x_i^+)$. Therefore, to prove $R_\epsilon$ is convex, it is enough to show that $\min\{R_\epsilon^{(2)}(x), R_\epsilon^{(2)}(x_i^+), R_\epsilon^{(2)}(x_{i+1}^-) : x \in (x_i, x_{i+1})\} \geq 0$ for each $i \in \mathbb{N}_{N-1}$. Let the shape parameters satisfy the assumptions given in the statement. Now, differentiating twice $R_\epsilon$ given in (5), we obtain

$$R_\epsilon^{(2)}(\mathcal{L}_i(x)) = \frac{\sum_{k=0}^5 \mathring{D}_{ik}(1-\theta)^{5-k}\theta^k}{h_i^* Q_i^3(\theta)}, \quad i \in \mathbb{N}_{N-1},$$

where

$$\mathring{D}_{i0} = 4\eta_i\sigma_i^3(\Delta_i - d_{i+\epsilon_i}) + 2\eta_i\sigma_i^2(\Delta_i - d_{i+1-\epsilon_i}),$$
$$\mathring{D}_{i1} = 8\eta_i\sigma_i^3(\Delta_i - d_{i+\epsilon_i}) + 4\eta_i\sigma_i^2(\Delta_i - d_{i+1-\epsilon_i}) + 6\eta_i\sigma_i^2(\Delta_i - d_{i+\epsilon_i}),$$
$$\mathring{D}_{i2} = 6\eta_i^2\sigma_i(d_{i+1-\epsilon_i} - \Delta_i) + 4\eta_i\sigma_i^3(\Delta_i - d_{i+\epsilon_i}) + 12\eta_i\sigma_i^2(\Delta_i - d_{i+\epsilon_i}) + 2\eta_i\sigma_i^2(\Delta_i - d_{i+1-\epsilon_i}),$$
$$\mathring{D}_{i3} = 6\eta_i\sigma_i^2(\Delta_i - d_{i+\epsilon_i}) + 4\eta_i^3\sigma_i(d_{i+1-\epsilon_i} - \Delta_i) + 12\eta_i^2\sigma_i(d_{i+1-\epsilon_i} - \Delta_i) + 2\eta_i^2\sigma_i(d_{i+\epsilon_i} - \Delta_i),$$
$$\mathring{D}_{i4} = 8\eta_i^3\sigma_i(d_{i+1-\epsilon_i} - \Delta_i) + 4\eta_i^2\sigma_i(d_{i+\epsilon_i} - \Delta_i) + 6\eta_i^2\sigma_i(d_{i+1-\epsilon_i} - \Delta_i),$$
$$\mathring{D}_{i5} = 4\eta_i^3\sigma_i(d_{i+1-\epsilon_i} - \Delta_i) + 2\eta_i^2\sigma_i(d_{i+\epsilon_i} - \Delta_i).$$

If $\epsilon_i = 0$, then $h_i^* > 0$, and the given assumptions on the shape parameters confirm that $Q_i^3 > 0$ and $\mathring{D}_{ik} \geq 0$ for each $k = 0, 1, \ldots, 5$. Therefore, $R_\epsilon^{(2)}(\mathcal{L}_i(x)) \geq 0$ for all $x \in I$. Similarly, if $\epsilon_i = 1$, then $h_i^* < 0$, and the given assumptions on the shape parameters confirm that $Q_i^3 > 0$ and $\mathring{D}_{ik} \leq 0$ for each $k = 0, 1, \ldots, 5$. Therefore, $R_\epsilon^{(2)}(\mathcal{L}(x)) \geq 0$ for all $x \in I$ and for each $i = 1, 2, \ldots, N-1$. Hence, $R_\epsilon$ is convex. $\square$

**Remark 1.** *When $\epsilon_i = 0$ for all $i \in \mathbb{N}_{N-1}$, the sufficient conditions given in Theorem 5 reduce to sufficient conditions:*

$$\sigma_i > \max\left\{0, \frac{d_{i+1} - \Delta_i}{2(\Delta_i - d_i)}\right\}, \quad \eta_i > \max\left\{0, \frac{\Delta_i - d_i}{2(d_{i+1} - \Delta_i)}\right\},$$

*for a classical rational cubic interpolant defined in [33] to be convex for given convex interpolation data.*

Now, using the convexity of $R_\epsilon$, we will show that $R_\epsilon^\alpha$ is also convex if the scaling and shape parameters are restricted suitably. We have

$$R_\epsilon^\alpha(\mathcal{L}_i(x)) = \alpha_i R_\epsilon^\alpha(x) + \frac{R_i^{**}(\theta)}{Q_i(\theta)}, \tag{10}$$

where

$$R_i^{**}(\theta) = \sum_{k=0}^3 C_{ik}(1-\theta)^{3-k}\theta^k, \quad C_{i0} = \sigma_i(y_{i+\epsilon_i} - \alpha_i y_1),$$
$$C_{i1} = (2\sigma_i\eta_i + \sigma_i)(y_{i+\epsilon_i} - \alpha_i y_1) + \sigma_i(h_i^* d_{i+\epsilon_i} - \alpha_i|I|d_1),$$
$$C_{i2} = (2\sigma_i\eta_i + \eta_i)(y_{i+1-\epsilon_i} - \alpha_i y_N)) - \eta_i(h_i^* d_{i+1-\epsilon_i} - \alpha_i|I|d_N),$$
$$C_{i3} = \eta_i(y_{i+1-\epsilon_i} - \alpha_i y_N).$$

**Theorem 6.** *Let $\{(x_i, y_i, d_i) : i = 1, 2, \ldots, N\}$ be a convex Hermite dataset, where $d_1 < \Delta_1 < \ldots < d_i < \Delta_i < d_{i+1} < \Delta_{i+1} < \ldots < \Delta_{N-1} < d_N$. Let $\epsilon \in \{0, 1\}^{N-1}$ be fixed and the shape parameters be chosen as*

$$\sigma_i > \max\left\{0, \frac{d_{i+1-\epsilon_i} - \Delta_i}{2(\Delta_i - d_{i+\epsilon_i})}\right\}, \quad \eta_i > \max\left\{0, \frac{\Delta_i - d_{i+\epsilon_i}}{2(d_{i+1-\epsilon_i} - \Delta_i)}\right\}, \quad i \in \mathbb{N}_{N-1}.$$

*If we choose scaling factors such that*

$$0 \leq \alpha_i < a_i^2 \min\left\{1, \frac{\nu''}{\mathcal{V}_{i,2}}\right\}, \quad i \in \mathbb{N}_{N-1},$$

*where* $\nu'' := \min\limits_{x \in I}\{R_\epsilon^{(2)}(x^+), R_\epsilon^{(2)}(x^-)\}$ *and* $\mathcal{V}_{i,2} > \max\{0, \max\{\mathcal{B}_i^{(2)} : x \in I\}\}, i \in \mathbb{N}_{N-1},$ *then the corresponding $C^1$-RCS ZFIF $R_\epsilon^\alpha$ will be convex on I.*

**Proof.** From the construction (5), it is obvious that the second derivative of $R_\epsilon$ exists on each $x \in (x_i, x_{i+1}), i \in \mathbb{N}_{N-1}$, and also, $R_\epsilon^2(x_i^+)$ and $R_\epsilon^2(x_{i+1}^-)$ exist. However, at the endpoints of the subintervals, it can happen that $R_\epsilon^2(x_i^-) \neq R_\epsilon^2(x_i^+)$. Therefore, using the given assumptions on the shape parameters in Theorem 5, we have $\min\{R_\epsilon^2(x), R_\epsilon^2(x_i^+), R_\epsilon^2(x_{i+1}^-) : x \in (x_i, x_{i+1})\} \geq 0$. Thus, $\nu'' > 0$. For each $i \in \mathbb{N}_{N-1}, \mathcal{B}_i$ is infinitely differentiable on $I$.

Let the zipper IFS parameters satisfy the assumptions given in the statement. To prove $R_\epsilon^\alpha$ is convex, it is enough to prove that $(R_\epsilon^\alpha)^{(2)}(x^+)$ or $(R_\epsilon^\alpha)^{(2)}(x^-)$ exists and is non-negative (possibly $+\infty$) for all $x \in (x_1, x_N)$ (see, Proposition 1). Let $0 \leq \alpha_i < a_i^2$. From (9), informally, we can write

$$(R_\epsilon^\alpha)^{(2)}(\mathcal{L}_i(x)) = R_\epsilon^{(2)}(\mathcal{L}_i(x)) + \frac{\alpha_i}{a_i^2}(R_\epsilon^\alpha)^{(2)}(x) - \frac{\alpha_i}{a_i^2}\mathcal{B}_i^{(2)}(x), \quad i \in \mathbb{N}_{N-1}. \tag{11}$$

First, we will prove that the assumptions on parameters imply $(R_\epsilon^\alpha)^{(2)}(x_1^+) \geq 0$ and $(R_\epsilon^\alpha)^{(2)}(x_N^-) \geq 0$.

Case 1 : Let $\epsilon_1 = 0$ and $\epsilon_{N-1} = 0$.

Taking $i = 1$ and $x = x_1$ in (11), we obtain

$$(R_\epsilon^\alpha)^{(2)}(x_1^+) = \left(1 - \frac{\alpha_1}{a_1^2}\right)^{-1}[R_\epsilon^{(2)}(x_1^+) - \frac{\alpha_1}{a_1^2}\mathcal{B}_1^{(2)}(x_1^+)].$$

Now, $R_\epsilon^{(2)}(x_1^+) - \frac{\alpha_1}{a_1^2}\mathcal{B}_1^{(2)}(x_1^+) \geq \nu'' - \frac{\alpha_1}{a_1^2}\mathcal{V}_{1,2}$. Thus, $\alpha_1 \leq \frac{a_1^2 \nu''}{\mathcal{V}_{1,2}}$ implies $(R_\epsilon^\alpha)^{(2)}(x_1^+) \geq 0$. Taking $i = N - 1$ and $x = x_N$, we have

$$(R_\epsilon^\alpha)^{(2)}(x_N^-) = \left(1 - \frac{\alpha_{N-1}}{a_{N-1}^2}\right)^{-1}[R_\epsilon^{(2)}(x_N^-) - \frac{\alpha_{N-1}}{a_{N-1}^2}\mathcal{B}_{N-1}^{(2)}(x_N^-)].$$

Similarly, we can obtain $(R_\epsilon^\alpha)^{(2)}(x_N^-) \geq 0$ if $\alpha_{N-1} \leq \frac{a_{N-1}^2 \nu''}{\mathcal{V}_{N-1,2}}$.

Case 2 : Let $\epsilon_1 = 1$ and $\epsilon_{N-1} = 1$.

From (11), we obtain

$$(R_\epsilon^\alpha)^{(2)}(x_1^+) = R_\epsilon^{(2)}(x_1^+) + \frac{\alpha_1}{a_1^2}(R_\epsilon^\alpha)^{(2)}(x_N^-) - \frac{\alpha_1}{a_1^2}\mathcal{B}_1^{(2)}(x_N^-), \tag{12}$$

$$(R_\epsilon^\alpha)^{(2)}(x_N^-) = R_\epsilon^{(2)}(x_N^-) + \frac{\alpha_{N-1}}{a_{N-1}^2}(R_\epsilon^\alpha)^{(2)}(x_1^+) - \frac{\alpha_{N-1}}{a_{N-1}^2}\mathcal{B}_{N-1}^{(2)}(x_1^+). \tag{13}$$

Adding (12) and (13), we have

$$\left(1 - \frac{\alpha_{N-1}}{a_{N-1}^2}\right)(R_\epsilon^\alpha)^{(2)}(x_1^+) + \left(1 - \frac{\alpha_1}{a_1^2}\right)(R_\epsilon^\alpha)^{(2)}(x_N^-) =$$

$$\left[R_\epsilon^{(2)}(x_1^+) - \frac{\alpha_1}{a_1^2}\mathcal{B}_1^{(2)}(x_N^-)\right] + \left[R_\epsilon^{(2)}(x_N^-) - \frac{\alpha_{N-1}}{a_{N-1}^2}\mathcal{B}_{N-1}^{(2)}(x_1^+)\right].$$

If $\alpha_1 \leq \frac{a_1^2 v''}{\mathcal{V}_{1,2}}$ and $\alpha_{N-1} \leq \frac{a_{N-1}^2 v''}{\mathcal{V}_{N-1,2}}$, then

$$\left(1 - \frac{\alpha_{N-1}}{a_{N-1}^2}\right)(R_\epsilon^\alpha)^{(2)}(x_1^+) + \left(1 - \frac{\alpha_1}{a_1^2}\right)(R_\epsilon^\alpha)^{(2)}(x_N^-) \geq 0.$$

This implies either $(R_\epsilon^\alpha)^{(2)}(x_1^+) \geq 0$ or $(R_\epsilon^\alpha)^{(2)}(x_N^-) \geq 0$. Therefore, if $(R_\epsilon^\alpha)^{(2)}(x_1^+) \geq 0$, then (13) implies

$$(R_\epsilon^\alpha)^{(2)}(x_N^-) \geq R_\epsilon^{(2)}(x_N^-) - \frac{\alpha_{N-1}}{a_{N-1}^2}\mathcal{B}_{N-1}^{(2)}(x_1^+) \geq 0.$$

Similarly, we can prove that if $(R_\epsilon^\alpha)^{(2)}(x_N^-) \geq 0$, then $(R_\epsilon^\alpha)^{(2)}(x_1^+) \geq 0$.

Case 3 : Let $\epsilon_1 = 0$ and $\epsilon_{N-1} = 1$.

We have

$$(R_\epsilon^\alpha)^{(2)}(x_1^+) = \left(1 - \frac{\alpha_1}{a_1^2}\right)^{-1}\left[R_\epsilon^{(2)}(x_1^+) - \frac{\alpha_1}{a_1^2}\mathcal{B}_1^{(2)}(x_1^+)\right], \tag{14}$$

$$(R_\epsilon^\alpha)^{(2)}(x_N^-) = R_\epsilon^{(2)}(x_N^-) + \frac{\alpha_{N-1}}{a_{N-1}^2}(R_\epsilon^\alpha)^{(2)}(x_1^+) - \frac{\alpha_{N-1}}{a_{N-1}^2}\mathcal{B}_{N-1}^{(2)}(x_1^+). \tag{15}$$

Using the steps from Case 1 and Case 2, we can conclude that $(R_\epsilon^\alpha)^{(2)}(x_1^+) \geq 0$ and $(R_\epsilon^\alpha)^{(2)}(x_N^-) \geq 0$ if $\alpha_1 \leq \frac{a_1^2 v''}{\mathcal{V}_{1,2}}$ and $\alpha_{N-1} \leq \frac{a_{N-1}^2 v''}{\mathcal{V}_{N-1,2}}$.

Case 4 : Let $\epsilon_1 = 1$ and $\epsilon_{N-1} = 0$. This case is similar to the previous case.

Thus, we have

$$(R_\epsilon^\alpha)^{(2)}(x_1^+) \geq 0 \text{ and } (R_\epsilon^\alpha)^{(2)}(x_N^-) \geq 0. \tag{16}$$

Now, we are moving to the other knots. For a fixed $i \in \mathbb{N}_{N-1}$, if $\epsilon_i = 0$, then (11) gives

$$(R_\epsilon^\alpha)^{(2)}(x_i^+) = R_\epsilon^{(2)}(x_i^+) + \frac{\alpha_i}{a_i^2}(R_\epsilon^\alpha)^{(2)}(x_1^+) - \frac{\alpha_i}{a_i^2}\mathcal{B}_i^{(2)}(x_1^+). \tag{17}$$

Now, using $(R_\epsilon^\alpha)^{(2)}(x_1^+) \geq 0$ and $\alpha_i \leq \frac{a_i^2 v''}{\mathcal{V}_{i,2}}$, we have $(R_\epsilon^\alpha)^{(2)}(x_i^+) \geq 0$.

If $\epsilon_i = 1$, we have

$$(R_\epsilon^\alpha)^{(2)}(x_{i+1}^-) = R_\epsilon^{(2)}(x_{i+1}^-) + \frac{\alpha_i}{a_i^2}(R_\epsilon^\alpha)^{(2)}(x_1^+) - \frac{\alpha_i}{a_i^2}\mathcal{B}_i^{(2)}(x_1^+).$$

Similarly, we can obtain $(R_\epsilon^\alpha)^{(2)}(x_{i+1}^-) \geq 0$ using the assumptions on the zipper IFS parameters. Hence, the condition

$$(R_\epsilon^\alpha)^{(2)}(x_1^+) \geq 0 \text{ and } \alpha_i \leq \frac{a_i^2 v''}{\mathcal{V}_{i,2}} \quad \Rightarrow \quad \begin{cases} (R_\epsilon^\alpha)^{(2)}(x_i^+) \geq 0 & \text{if } \epsilon_i = 0, \\ (R_\epsilon^\alpha)^{(2)}(x_{i+1}^-) \geq 0 & \text{if } \epsilon_i = 1 \text{ for } i \in \mathbb{N}_{N-1}. \end{cases} \tag{18}$$

Similarly,

$$(R_\epsilon^\alpha)^{(2)}(x_N^-) \geq 0 \text{ and } \alpha_i \leq \frac{a_i^2 v''}{\mathcal{V}_{i,2}} \quad \Rightarrow \quad \begin{cases} (R_\epsilon^\alpha)^{(2)}(x_i^+) \geq 0 & \text{if } \epsilon_i = 1, \\ (R_\epsilon^\alpha)^{(2)}(x_{i+1}^-) \geq 0 & \text{if } \epsilon_i = 0 \text{ for } i \in \mathbb{N}_{N-1}. \end{cases} \tag{19}$$

Therefore, using the given assumptions on the zipper IFS parameters, we have

$$\min\left\{(R_\epsilon^\alpha)^{(2)}(x_1^+), (R_\epsilon^\alpha)^{(2)}(x_N^-), (R_\epsilon^\alpha)^{(2)}(x_i^-), (R_\epsilon^\alpha)^{(2)}(x_i^+) : i = 2, 3, \ldots, N-1\right\} \geq 0.$$

Now, we are moving to the next iteration. If $\epsilon_i = 0$, we have

$$(R_{\hat{\epsilon}}^{\alpha})^{(2)}(\mathcal{L}_i(x_j)^+) = R_{\hat{\epsilon}}^{(2)}(\mathcal{L}_i(x_j)^+) + \frac{\alpha_i}{a_i^2}(R_{\hat{\epsilon}}^{\alpha})^{(2)}(x_j^+) - \frac{\alpha_i}{a_i^2}\mathcal{B}_i^{(2)}(x_j^+),$$

$$(R_{\hat{\epsilon}}^{\alpha})^{(2)}(\mathcal{L}_i(x_j)^-) = R_{\hat{\epsilon}}^{(2)}(\mathcal{L}_i(x_j)^-) + \frac{\alpha_i}{a_i^2}(R_{\hat{\epsilon}}^{\alpha})^{(2)}(x_j^-) - \frac{\alpha_i}{a_i^2}\mathcal{B}_i^{(2)}(x_j^-), j = 2, 3, \ldots, N-1.$$

If $\epsilon_i = 1$, we have

$$(R_{\hat{\epsilon}}^{\alpha})^{(2)}(\mathcal{L}_i(x_j)^+) = R_{\hat{\epsilon}}^{(2)}(\mathcal{L}_i(x_j)^+) + \frac{\alpha_i}{a_i^2}(R_{\hat{\epsilon}}^{\alpha})^{(2)}(x_j^-) - \frac{\alpha_i}{a_i^2}\mathcal{B}_i^{(2)}(x_j^-),$$

$$(R_{\hat{\epsilon}}^{\alpha})^{(2)}(\mathcal{L}_i(x_j)^-) = R_{\hat{\epsilon}}^{(2)}(\mathcal{L}_i(x_j)^-) + \frac{\alpha_i}{a_i^2}(R_{\hat{\epsilon}}^{\alpha})^{(2)}(x_j^+) - \frac{\alpha_i}{a_i^2}\mathcal{B}_i^{(2)}(x_j^+), j = 2, 3, \ldots, N-1.$$

Using a similar procedure, we have $\min_{i \in \mathbb{N}_{N-1}}\{(R_{\hat{\epsilon}}^{\alpha})^{(2)}(\mathcal{L}_i(x_j)^+), (R_{\hat{\epsilon}}^{\alpha})^{(2)}(\mathcal{L}_i(x_j)^-) : j \in \mathbb{N}_N\} \geq 0$, provided the parameters satisfy the given assumptions. Similarly, $(R_{\hat{\epsilon}}^{\alpha})^2(x^+)$ and $(R_{\hat{\epsilon}}^{\alpha})^2(x^-)$ exist, and are non-negative on the new points generated in the next iteration. Since $(R_{\hat{\epsilon}}^{\alpha})^{(2)}$ is determined iteratively, we can conclude that $R_{\hat{\epsilon}}^{\alpha}$ is convex. □

**Example 2.** *Consider a univariate Hermite interpolation dataset*

$$\{(-4, 0, 0.25), (-2, 1, 1), (0, 4, 1.8), (2, 8, 3.2), (4, 15, 4)\}.$$

*Clearly, the given Hermite dataset is convex and satisfies $d_1 < \Delta_1 < d_2 < \Delta_2 < d_3 < \Delta_3 < d_4 < \Delta_4 < d_5$. Using the parameters given in Table 2 and signature $\epsilon = (1, 1, 1, 0)$, we plotted the proposed interpolants in Figure 2a–c.*

*We chose random parameters for plotting Figure 2a, and the corresponding RCS ZFIF is not convex, which can be seen from its first derivative plotted in Figure 2d, as it is not monotonically increasing, or it can be seen from its double derivative plotted in Figure 2g, as it takes some negative values.*

*We used sufficient conditions provided in Theorems 5 and 6 to plot the RCS ZFIF in Figure 2b and zipper RCS in Figure 2c, respectively. From their figures or the plot of their second derivatives in Figure 2h,i, it can be observed that they preserve the convexity of the given data. As the magnitude of each scaling factor is close to zero, the difference between the RCS ZFIF plotted in Figure 2b and zipper RCS plotted in Figure 2c cannot be seen much from their plots, but it can be easily observed from their second derivatives. In addition, one can observe that the second derivative of the RCS ZFIF plotted in Figure 2h does not exist on more points than the second derivative of the zipper RCS plotted in Figure 2i.*

**Table 2.** Shape parameters and scaling vectors for Figure 2.

| Figure 2 | $\sigma$ | $\eta$ | $\alpha$ |
|---|---|---|---|
| (a) | $(0.03, 0.01, 0.1, 4)$ | $(1, 0.02, 1, 0.01)$ | $(-0.02, 0.02, -0.03, -0.03)$ |
| (b) | $(1, 2, 5, 2)$ | $(2, 4, 5, 3)$ | $(0.01, 0, 0.01, 0.01)$ |
| (c) | $(1, 2, 5, 2)$ | $(2, 4, 5, 3)$ | $--$ |

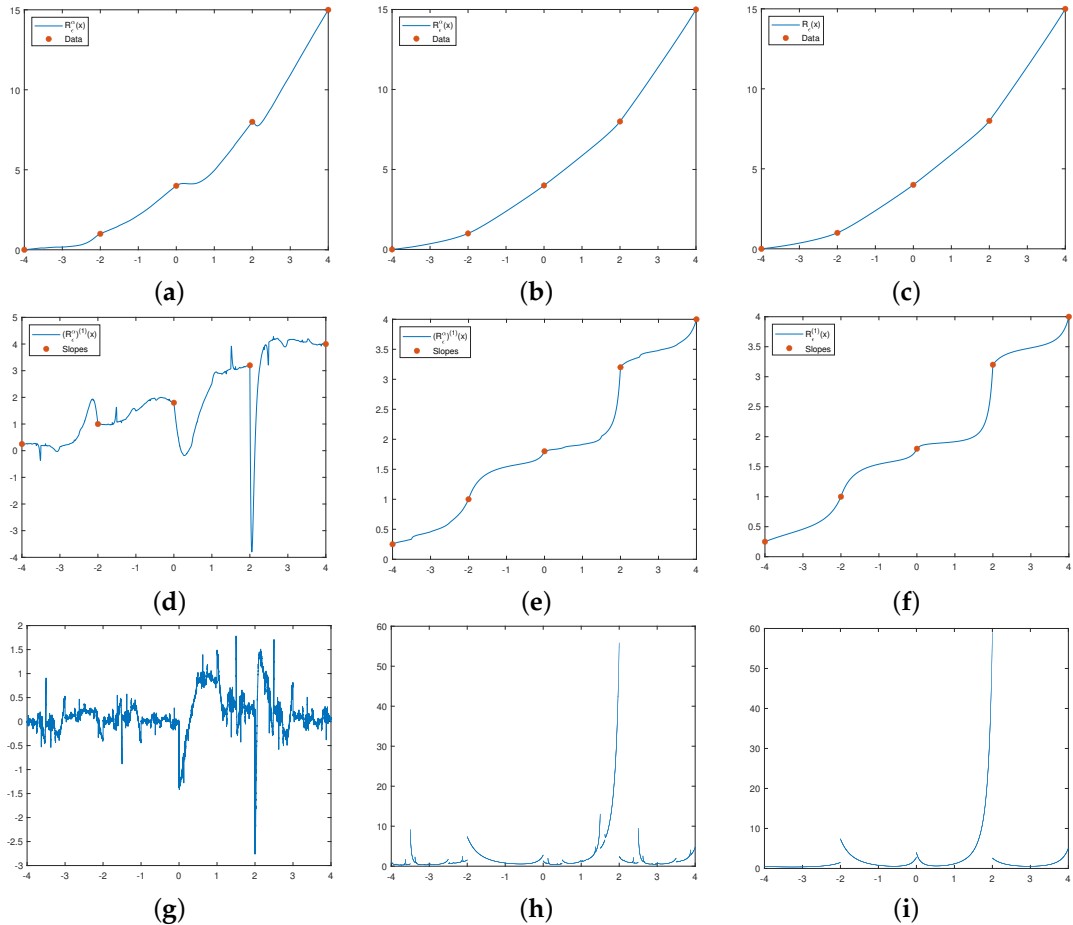

**Figure 2.** Verification of convexity by RCS ZFIFs. (**a**) Nonconvex RCS ZFIF. (**b**) Convex RCS ZFIF. (**c**) Convex zipper RCS. (**d**) The first derivative of the RCS ZFIF plotted in Figure 2a. (**e**) The first derivative of the RCS ZFIF plotted in Figure 2b. (**f**) The first derivative of the RCS ZFIF plotted in Figure 2c. (**g**) The second derivative of the zipper RCS plotted in Figure 2a. (**h**) The second derivative of the zipper RCS plotted in Figure 2b. (**i**) The second derivative of the zipper RCS plotted in Figure 2c.

## 5. Construction of Bicubic Partially Blended RCZFIS

In this section, we will generate zipper fractal interpolation surfaces for given Hermite bivariate data $\Gamma := \{(x_i, y_j, z_{i,j}, z_{i,j}^x, z_{i,j}^y) : i \in \mathbb{N}_N, \ j \in \mathbb{N}_M\}$, where $x_1 < x_2 < \ldots < x_N$, $y_1 < y_2 < \ldots < y_M$, and $z_{i,j}^x$ is $x$-partial and $z_{i,j}^y$ is $y$-partial at the point $(x_i, y_j)$. First, we split the given Hermite bivariate or surface data into univariate Hermite datasets along the $x$-axis and the $y$-axis, i.e., along each $j$-th grid line parallel to the $x$-axis and along each $i$-th grid line parallel to the $y$-axis, see Figure 3. Thus, we have $M$ number of Hermite datasets $\Gamma_{y_j} := \{(x_i, z_{i,j}, z_{i,j}^x) : i \in \mathbb{N}_N\}$ along the $x$-axis and $N$ number of Hermite datasets $\Gamma_{x_i} := \{(y_j, z_{i,j}, z_{i,j}^y) : j \in \mathbb{N}_M\}$ along the $y$-axis. Then, we construct RCS ZFIFs using these univariate Hermite datasets. We generate a zipper fractal interpolation surface with the Coons patch technique [28] using these RCS ZFIFs as the boundary curves and blending them with two cubic blending functions. Here, we obtain the advantage of using signature vectors as we can obtain $M \times N \times 2^{N+M-2}$ different zipper fractal interpolation surfaces by varying the values of signature vectors only.

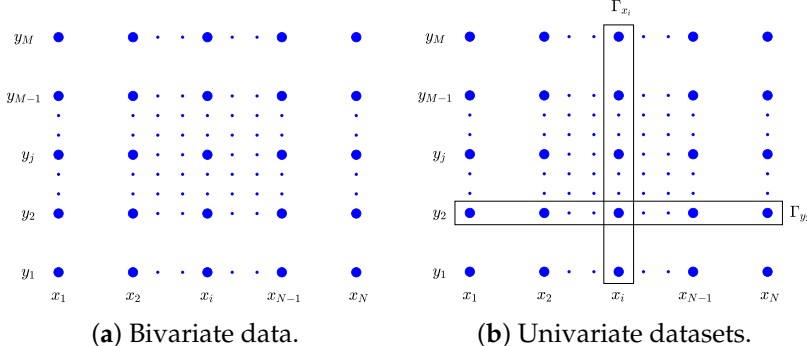

**(a)** Bivariate data. **(b)** Univariate datasets.

**Figure 3.** Splitting bivariate data into univariate datasets.

For $i \in \mathbb{N}_{N-1}$ and $j \in \mathbb{N}_{M-1}$, denote $I := [x_1, x_N]$, $I_i := [x_i, x_{i+1}]$, $J := [y_1, y_M]$, $J_j := [y_j, y_{j+1}]$, $R_{i,j} := I_i \times J_j$, $R := I \times J = \cup_{i=1}^{N-1} \cup_{j=1}^{M-1} R_{i,j}$, $h_i := x_{i+1} - x_i$, $h := \max\{h_i : i \in \mathbb{N}_{N-1}\}$, $\overline{h}_j := y_{j+1} - y_j$, $\overline{h} := \max\{\overline{h}_j : j \in \mathbb{N}_{M-1}\}$, $\Delta_i^j := \frac{z_{i+1,j}-z_{i,j}}{x_{i+1}-x_i}$, and $\overline{\Delta}_j^i := \frac{z_{i,j+1}-z_{i,j}}{y_{j+1}-y_j}$. For fixed $j \in \mathbb{N}_M$, consider the univariate Hermite data $\Gamma_{y_j}$. Let the signature corresponding to the dataset $\Gamma_{y_j}$ denoted by $\epsilon^j = (\epsilon_1^j, \epsilon_2^j, \ldots, \epsilon_{N-1}^j) \in \{0,1\}^{N-1}$ be fixed. For each $i \in \mathbb{N}_{N-1}$, let $\mathcal{L}_i^j : [x_1, x_N] \to [x_i, x_{i+1}]$ be a contractive homeomorphism such that $\mathcal{L}_i^j(x) = a_i^j x + b_i^j$ satisfying $\mathcal{L}_i^j(x_1) = x_{i+\epsilon_i^j}$ and $\mathcal{L}_i^j(x_N) = x_{i+1-\epsilon_i^j}$, $\alpha_i^j$ be a scaling factor such that $|\alpha_i^j| < |a_i^j|$, $\sigma_i^j$ and $\eta_i^j > 0$ be the shape parameters, $h_i^j := x_{i+1-\epsilon_i^j} - x_{i+\epsilon_i^j}$, and $\mathcal{B}_i^j$ be a base function. Using these notations in Section 3, the RCS ZFIF corresponding to the univariate Hermite data $\Gamma_{y_j}$ denoted by $\mathcal{N}_{\epsilon^j}^{\alpha^j}$ can be written as

$$\mathcal{N}_{\epsilon^j}^{\alpha^j}(x) = \mathcal{N}_{\epsilon^j}(x) + \alpha_i^j \left( \mathcal{N}_{\epsilon^j}^{\alpha^j}((\mathcal{L}_i^j)^{-1}(x)) - \mathcal{B}_i^j((\mathcal{L}_i^j)^{-1}(x)) \right), \ x \in I_i, \ i \in \mathbb{N}_{N-1}, \quad (20)$$

where $\mathcal{N}_{\epsilon^j}$ is the univariate Hermite zipper RCS that interpolates the data $\Gamma_{y_j}$. Thus,

$$\mathcal{N}_{\epsilon^j}^{\alpha^j}(x_i) = \mathcal{N}_{\epsilon^j}(x_i) = z_{i,j}, \quad \frac{d\mathcal{N}_{\epsilon^j}^{\alpha^j}(x_i)}{dx} = \frac{d\mathcal{N}_{\epsilon^j}(x_i)}{dx} = z_{i,j}^x, \ \forall j \in \mathbb{N}_M, \ \forall i \in \mathbb{N}_N. \quad (21)$$

By fixing the signature $\epsilon^j$ for each univariate dataset $\Gamma_{y_j}$, $j \in \mathbb{N}_M$, we can generate a RCS ZFIF, which interpolates $\Gamma_{y_j}$, i.e., we can construct $M$ number of RCS ZFIFs along the $x$-axis.

Similarly, we construct RCS ZFIFs along the $y$-axis. For fixed $i \in \mathbb{N}_N$, consider the univariate Hermite data $\Gamma_{x_i} = \{(y_j, z_{i,j}, z_{i,j}^y) : j \in \mathbb{N}_M\}$. Let the signature corresponding to the data $\Gamma_{x_i}$ denoted by $\overline{\epsilon}^i = (\overline{\epsilon}_1^i, \overline{\epsilon}_2^i, \ldots, \overline{\epsilon}_{M-1}^i) \in \{0,1\}^{M-1}$ be fixed. For each $j \in \mathbb{N}_{M-1}$, let $\overline{\mathcal{L}}_j^i : [y_1, y_M] \to [y_j, y_{j+1}]$ be a contractive homeomorphism such that $\overline{\mathcal{L}}_j^i(y) = \overline{a}_j^i y + \overline{b}_j^i$ satisfying $\overline{\mathcal{L}}_j^i(y_1) = y_{j+\overline{\epsilon}_j^i}$ and $\overline{\mathcal{L}}_j^i(y_M) = y_{j+1-\overline{\epsilon}_j^i}$, $\overline{\alpha}_j^i$ be a scaling factor such that $|\overline{\alpha}_j^i| < |\overline{a}_j^i|$, $\overline{\sigma}_j^i$ and $\overline{\eta}_j^i > 0$ be the shape parameters, $\overline{h}_j^i := y_{j+1-\overline{\epsilon}_j^i} - y_{i+\overline{\epsilon}_j^i}$, and $\overline{\mathcal{B}}_j^i$ be a base function.

Now, the RCS ZFIF corresponding to the univariate Hermite data $\Gamma_{x_i}$ denoted by $\overline{\mathcal{N}}_{\overline{\epsilon}^i}^{\overline{\alpha}^i}$ can be written as

$$\overline{\mathcal{N}}_{\overline{\epsilon}^i}^{\overline{\alpha}^i}(y) = \overline{\mathcal{N}}_{\overline{\epsilon}^i}(y) + \overline{\alpha}_j^i \left( \overline{\mathcal{N}}_{\overline{\epsilon}^i}^{\overline{\alpha}^i}((\overline{\mathcal{L}}_j^i)^{-1}(y)) - \overline{\mathcal{B}}_j^i((\overline{\mathcal{L}}_j^i)^{-1}(y)) \right), \ y \in J_j, \ j \in \mathbb{N}_{M-1}, \quad (22)$$

where $\overline{\mathcal{N}}_{\overline{\epsilon}^i}$ is the univariate Hermite zipper RCS that interpolates the data $\Gamma_{x_i}$. Thus,

$$\overline{\mathcal{N}}_{\overline{\epsilon}^i}^{\overline{\alpha}^i}(y_j) = \overline{\mathcal{N}}_{\overline{\epsilon}^i}(y_j) = z_{i,j}, \quad \frac{d\overline{\mathcal{N}}_{\overline{\epsilon}^i}^{\overline{\alpha}^i}(y_j)}{dy} = \frac{d\overline{\mathcal{N}}_{\overline{\epsilon}^i}(y_j)}{dy} = z_{i,j}^y, \ \forall i \in \mathbb{N}_N, \ \forall j \in \mathbb{N}_M. \quad (23)$$

Now, for the boundary of subrectangle $R_{i,j} = [x_i, x_{i+1}] \times [y_j, y_{j+1}]$, we have four curves (two along the $i$-th and $i+1$-th grid lines and two along the $j$-th and $j+1$-th grid lines) $\mathcal{N}_{\epsilon^j}^{\alpha^j}(x)\big|_{[x_i, x_{i+1}]}$, $\mathcal{N}_{\epsilon^{j+1}}^{\alpha^{j+1}}(x)\big|_{[x_i, x_{i+1}]}$, $\overline{\mathcal{N}}_{\overline{\epsilon}^i}^{\overline{\alpha}^i}(y)\big|_{[y_j, y_{j+1}]}$, and $\overline{\mathcal{N}}_{\overline{\epsilon}^{i+1}}^{\overline{\alpha}^{i+1}}(y)\big|_{[y_j, y_{j+1}]}$. Using the transfinite interpolation method via blending functions, we can define the surface $\mathcal{F}_\epsilon^\alpha$ of the form:

$$\mathcal{F}_\epsilon^\alpha(x,y) = \mathcal{F}_{\epsilon,1}^\alpha(x,y) + \mathcal{F}_{\epsilon,2}^\alpha(x,y) - \mathcal{F}_{\epsilon,3}^\alpha(x,y), \quad x \in I_i, \ y \in J_j, \ i \in \mathbb{N}_{N-1}, \ j \in \mathbb{N}_{M-1}, \quad (24)$$

where

$$\mathcal{F}_{\epsilon,1}^\alpha(x,y) := \Omega_0\big(\mu_i^*(x)\big)\overline{\mathcal{N}}_{\overline{\epsilon}^i}^{\overline{\alpha}_i}(y) + \Omega_1\big(\mu_i^*(x)\big)\overline{\mathcal{N}}_{\overline{\epsilon}^{i+1}}^{\overline{\alpha}_{i+1}}(y),$$

$$\mathcal{F}_{\epsilon,2}^\alpha(x,y) := \Omega_0\big(\overline{\mu}_j(y)\big)\mathcal{N}_{\epsilon^j}^{\alpha^j}(x) + \Omega_1\big(\overline{\mu}_j(y)\big)\mathcal{N}_{\epsilon^{j+1}}^{\alpha^{j+1}}(x),$$

$$\begin{aligned}\mathcal{F}_{\epsilon,3}^\alpha(x,y) := {} & \Omega_0\big(\mu_i^*(x)\big)\Omega_0\big(\overline{\mu}_j(y)\big)z_{i,j} + \Omega_0\big(\mu_i^*(x)\big)\Omega_1\big(\overline{\mu}_j(y)\big)z_{i,j+1} \\ & + \Omega_1\big(\mu_i^*(x)\big)\Omega_0\big(\overline{\mu}_j(y)\big)z_{i+1,j} + \Omega_1\big(\mu_i^*(x)\big)\Omega_1\big(\overline{\mu}_j(y)\big)z_{i+1,j+1},\end{aligned}$$

$$\mu_i^*(x) = \frac{x - x_i}{x_{i+1} - x_i}, \quad \overline{\mu}_j(y) = \frac{y - y_j}{y_{j+1} - y_j}, \quad \Omega_0(\mu) := (1-\mu)^2(1+2\mu), \quad \Omega_1(\mu) := \mu^2(3-2\mu).$$

In (24), $\mathcal{F}_{\epsilon,1}^\alpha$ and $\mathcal{F}_{\epsilon,2}^\alpha$ each generates a surface using two boundary curves by blending these curves with cubic functions. $\mathcal{F}_{\epsilon,1}^\alpha$ uses $\mathcal{N}_{\overline{\epsilon}^i}^{\overline{\alpha}_i}(y)$ and $\mathcal{N}_{\overline{\epsilon}^{i+1}}^{\overline{\alpha}_{i+1}}(y)$, and $\mathcal{F}_{\epsilon,2}^\alpha$ uses another two curves $\mathcal{N}_{\epsilon^j}^{\alpha^j}(x)$ and $\mathcal{N}_{\epsilon^{j+1}}^{\alpha^{j+1}}(x)$. Since we added the corners twice, we used $\mathcal{F}_{\epsilon,3}^\alpha$. These smooth cubic functions $\Omega_0$ and $\Omega_1$, satisfying

$$\begin{aligned}\Omega_0(0) = 1, \ \Omega_0(1) = 0, \ \Omega_0'(0) = 0, \ \Omega_0'(1) = 0, \\ \Omega_1(0) = 0, \ \Omega_1(1) = 1, \ \Omega_1'(0) = 0, \ \Omega_1'(1) = 0,\end{aligned} \qquad (25)$$

are called blending functions as they blend four different univariate functions together on the boundary to produce a well-defined surface.

**Theorem 7.** *Let $\{(x_i, y_j, z_{i,j}, z_{i,j}^x, z_{i,j}^y) : i \in \mathbb{N}_N, \ j \in \mathbb{N}_M\}$ be a given bivariate Hermite dataset. The bivariate function $\mathcal{F}_\epsilon^\alpha$ defined in (24) has the following properties:*

*(i)*   $\mathcal{F}_\epsilon^\alpha \in C^1(I \times J)$;

*(ii)*   $\mathcal{F}_\epsilon^\alpha$ *interpolates the given bivariate data, i.e.,*

$$\mathcal{F}_\epsilon^\alpha(x_i, y_j) = z_{i,j}, \quad \frac{\partial \mathcal{F}_\epsilon^\alpha}{\partial x}\bigg|_{(x_i, y_j)} = z_{i,j}^x, \quad \frac{\partial \mathcal{F}_\epsilon^\alpha}{\partial y}\bigg|_{(x_i, y_j)} = z_{i,j}^y, \ \forall i \in \mathbb{N}_N, \ \forall j \in \mathbb{N}_M;$$

*(iii)*   *If the given bivariate data $\{(x_i, y_j, z_{i,j}, z_{i,j}^x, z_{i,j}^y) : i \in \mathbb{N}_N, \ j \in \mathbb{N}_M\}$ is taken from a function $\Psi \in C^1(I \times J)$, then bivariate interpolant $\mathcal{F}_\epsilon^\alpha$ converges uniformly to $\Psi$ as $h \to 0$ and $\overline{h} \to 0$.*

Since boundary curves satisfying (20)–(23) and blending functions satisfying (25) are continuously differentiable functions, it is easy to observe that the generated surface $\mathcal{F}_\epsilon^\alpha$ holds properties (i) and (ii) given in Theorem 7. Part (iii) of this theorem can be proved in a similar manner as described in [15]. We call the bivariate interpolant $\mathcal{F}_\epsilon^\alpha$ a rational cubic zipper fractal interpolation surface (RCZFIS). For each $j \in \mathbb{N}_M$, we can generate $2^{N-1}$ number of different RCS ZFIFs corresponding to the univariate data $\Gamma_{y_j}$ by varying the signature vector $\epsilon^j$ only. Therefore, we can construct $M \times 2^{N-1}$ number of RCS ZFIFs along the $x$-axis. Similarly, we can construct $N \times 2^{M-1}$ number of RCS ZFIFs along the $y$-axis. Therefore, we can construct $N \times M \times 2^{N+M-2}$ number of different RCZFISs for the given bivariate data $\{(x_i, y_j, z_{i,j}, z_{i,j}^x, z_{i,j}^y) : i \in \mathbb{N}_N, \ j \in \mathbb{N}_M\}$ by varying the signature vectors only, where all the other parameters are the same. The proposed class of partially

blended rational cubic zipper fractal interpolation surfaces generalizes the existing class of partially blended rational cubic fractal interpolation surfaces and partially blended rational cubic interpolation surfaces. If we choose all the scaling factors to be zero, then the bicubic partially blended rational cubic ZFIS (24) reduces to the bicubic partially blended zipper rational cubic surface, and we denote it $\mathcal{F}_\epsilon$.

**Example 3.** *For the Hermite surface interpolation data points of the form $(x_i, y_j, z_{i,j}, z_{i,j}^x, z_{i,j}^y)$ given in Table 3, we constructed different RCZFISs using the different signature vectors given in Table 4 and the fixed shape parameters and scaling function vectors given in Table 5.*

*In Figure 4a,b, we plotted the RCS ZFIFs along the x-axis and the y-axis. We used them to plot RCZFIS 1 given in Figure 4c. Then, using different signature vectors only, we plotted RCZFIS 2, RCZFIS 3, and RCZFIS 4 in Figure 4d–f, respectively. One can observe that we obtain a different surface when we change the signature. Since for the given surface data of $4 \times 4$ points, we have $4 \times 4 \times 2^3 \times 2^3 = 1024$ choices to choose a signature vector, we can generate 1024 different RCZFISs by changing the values of signature vectors with the fixed sets of shape parameters and scaling functions.*

**Table 3.** Bivariate Hermite data.

| $\downarrow x/y \rightarrow$ | $-1$ | $-1/3$ | $1/3$ | $1$ |
|---|---|---|---|---|
| $-1$ | $(3, 1, -1)$ | $(-1, -2, 2)$ | $(-2, -12, 3)$ | $(5, 8, 1)$ |
| $-1/3$ | $(2, -4, -10)$ | $(-5, -2, 2)$ | $(3, 4, -3)$ | $(-9, 5, 4)$ |
| $1/3$ | $(-1, 5, 1)$ | $(4, -3, 3)$ | $(-5, -1, 5)$ | $(3, -4, -1)$ |
| $1$ | $(7, 3, 6)$ | $(-8, 2, 3)$ | $(1, -2, 4)$ | $(-3, 7, -2)$ |

**Table 4.** Signature vectors.

| | |
|---|---|
| RQZFIS 1: | $\epsilon^1 = \epsilon^2 = \epsilon^3 = \epsilon^4 = (1, 0, 0)$, <br> $\bar{\epsilon}^1 = \bar{\epsilon}^2 = \bar{\epsilon}^3 = \bar{\epsilon}^4 = (0, 1, 0)$. |
| RQZFIS 2: | $\epsilon^1 = \epsilon^2 = \epsilon^3 = \epsilon^4 = (0, 1, 1)$, <br> $\bar{\epsilon}^1 = \bar{\epsilon}^2 = \bar{\epsilon}^3 = \bar{\epsilon}^4 = (0, 1, 0)$. |
| RQZFIS 3: | $\epsilon^1 = \epsilon^2 = \epsilon^3 = \epsilon^4 = (0, 1, 1)$, <br> $\bar{\epsilon}^1 = \bar{\epsilon}^2 = \bar{\epsilon}^3 = \bar{\epsilon}^4 = (1, 0, 1)$. |
| RQZFIS 4: | $\epsilon^1 = \epsilon^2 = \epsilon^3 = \epsilon^4 = (0, 0, 0)$, <br> $\bar{\epsilon}^1 = \bar{\epsilon}^2 = \bar{\epsilon}^3 = \bar{\epsilon}^4 = (0, 0, 0)$. |

**Table 5.** Shape parameters and scaling function vectors.

$$\sigma^1 = \sigma^2 = \sigma^3 = \sigma^4 = (0.1, 0.2, 0.3),$$
$$\eta^1 = \eta^2 = \eta^3 = \eta^4 = (0.3, 0.6, 0.1),$$
$$\alpha^1 = \alpha^2 = \alpha^3 = \alpha^4 = (0.2, -0.3, 0.3),$$
$$\bar{\sigma}^1 = \bar{\sigma}^2 = \bar{\sigma}^3 = \bar{\sigma}^4 = (1, 0.2, 3),$$
$$\bar{\eta}^1 = \bar{\eta}^2 = \bar{\eta}^3 = \bar{\eta}^4 = (0.2, 0.6, 0.1),$$
$$\bar{\alpha}^1 = \bar{\alpha}^2 = \bar{\alpha}^3 = \bar{\alpha}^4 = (-0.3, 0.25, 0.3).$$

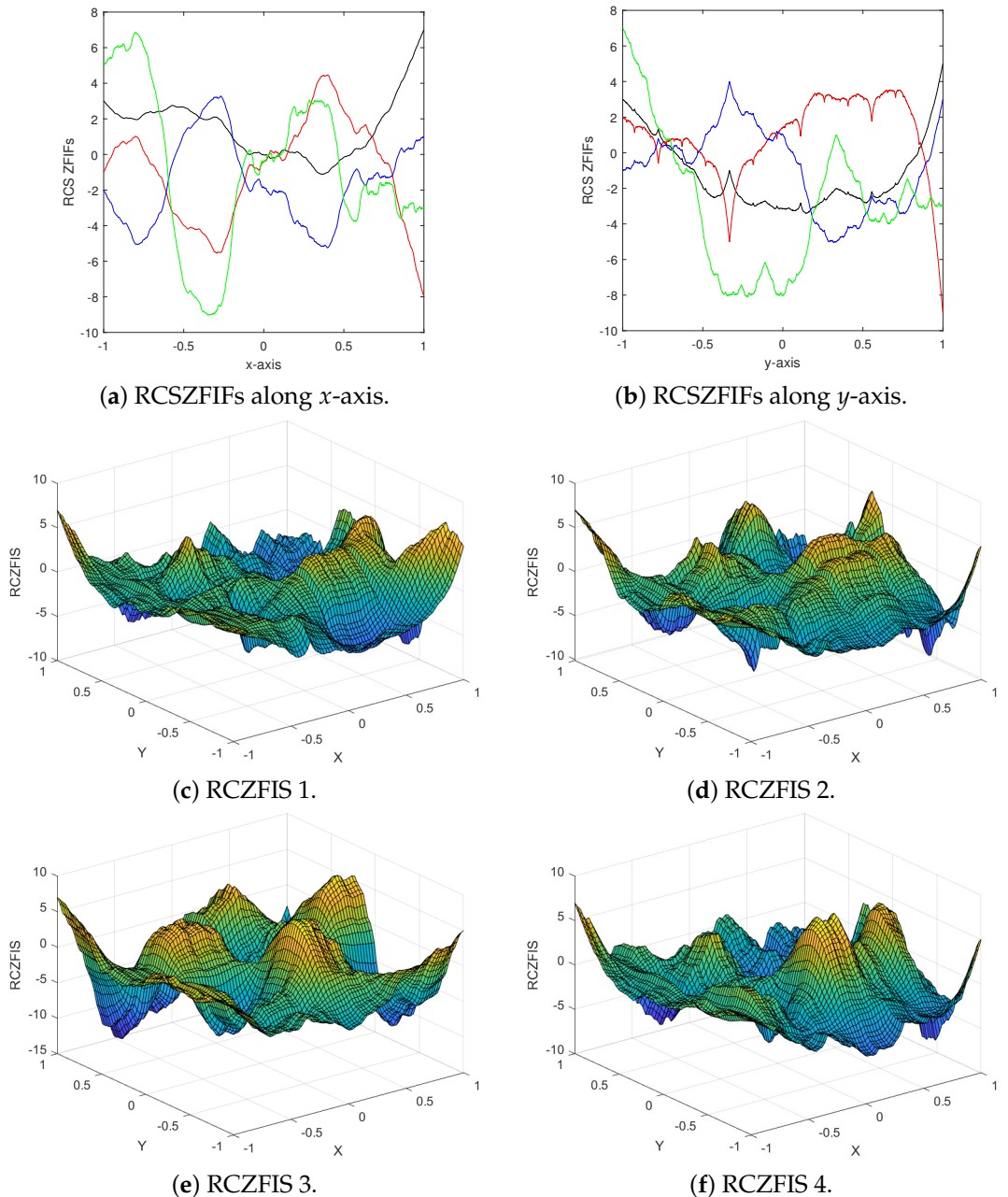

(**a**) RCSZFIFs along *x*-axis.

(**b**) RCSZFIFs along *y*-axis.

(**c**) RCZFIS 1.

(**d**) RCZFIS 2.

(**e**) RCZFIS 3.

(**f**) RCZFIS 4.

**Figure 4.** Zipper fractal network curves and surface interpolants.

## 6. Convexity-Preserving RCZFIS

In this section, we will generate convex RCZFISs for a given convex bivariate dataset. Casciola and Romani [29] observed that the bicubic partially blended surface inherits all the properties of the network of boundary curves.

Let $\Gamma = \{(x_i, y_i, z_{i,j}, z_{i,j}^x, z_{i,j}^y) : i \in \mathbb{N}_N, \ j \in \mathbb{N}_M\}$ be a convex surface data, where

$$\Delta_i^j < \Delta_{i+1}^j, \quad z_{i,j}^x < z_{i+1,j}^x, \quad z_{i,j}^x < \Delta_i^j < z_{i+1,j}^x; \quad i \in \mathbb{N}_{N-1}, j \in \mathbb{N}_M,$$

and

$$\overline{\Delta}_j^i < \overline{\Delta}_{j+1}^i, \quad z_{i,j}^y < z_{i,j+1}^y, \quad z_{i,j}^y < \overline{\Delta}_j^i < z_{i,j+1}^y; \quad i \in \mathbb{N}_N, j \in \mathbb{N}_{M-1}.$$

Therefore, the univariate Hermite datasets $\Gamma_{y_j}$ for all $j \in \mathbb{N}_M$ and $\Gamma_{x_i}$ for all $i \in \mathbb{N}_N$ are also convex. Now, with these notations in Theorem 6, we obtain the following results:

(i) For fixed $j \in \mathbb{N}_M$, $\mathcal{N}_{\epsilon^j}^{\alpha^j}$ is convex if

$$\sigma_i^j > \max\left\{0, \frac{z_{i+1-\epsilon_i^j,j}^x - \Delta_i^j}{2\left(\Delta_i^j - z_{i+\epsilon_i^j,j}^x\right)}\right\}, \quad \eta_i^j > \max\left\{0, \frac{\Delta_i^j - z_{i+\epsilon_i^j,j}^x}{2\left(z_{i+1-\epsilon_i^j,j}^x - \Delta_i^j\right)}\right\},$$

$$0 \le \alpha_i^j < (a_i^j)^2 \min\left\{1, \frac{(v'')^j}{\mathcal{V}_{i,2}^j}\right\}, \quad \forall i \in \mathbb{N}_{N-1},$$

(26)

where

$$(v'')^j := \max\left\{\mathcal{N}_{\epsilon^j}^{(2)}(x^+), \mathcal{N}_{\epsilon^j}^{(2)}(x^-) : x \in I\right\}, \quad \mathcal{V}_{i,2}^j > \max\left\{0, \max\left\{\mathcal{B}_i^{j^{(2)}}(x) : x \in I\right\}\right\};$$

(ii) For fixed $i \in \mathbb{N}_N$, $\overline{\mathcal{N}}_{\overline{\epsilon}^i}^{\overline{\alpha}^i}$ is convex if

$$\overline{\sigma}_j^i > \max\left\{0, \frac{z_{i,j+1-\overline{\epsilon}_j^i}^y - \overline{\Delta}_j^i}{2\left(\overline{\Delta}_j^i - z_{i,j+\overline{\epsilon}_j^i}^y\right)}\right\}, \quad \overline{\eta}_j^i > \max\left\{0, \frac{\overline{\Delta}_j^i - z_{i,j+\overline{\epsilon}_j^i}^y}{2\left(z_{i,j+1-\overline{\epsilon}_j^i}^y - \overline{\Delta}_j^i\right)}\right\},$$

$$0 \le \overline{\alpha}_j^i < (\overline{a}_j^i)^2 \min\left\{1, \frac{(\overline{v''})^i}{\overline{\mathcal{V}}_{j,2}^i}\right\}, \quad \forall j \in \mathbb{N}_{M-1},$$

(27)

where

$$(\overline{v''})^i := \max\left\{\overline{\mathcal{N}}_{\overline{\epsilon}^i}^{(2)}(y^+), \overline{\mathcal{N}}_{\overline{\epsilon}^i}^{(2)}(y^-) : y \in J\right\}, \quad \overline{\mathcal{V}}_{j,2}^i > \max\left\{0, \max\left\{\overline{\mathcal{B}}_j^{i^{(2)}}(y) : y \in J\right\}\right\}.$$

Thus, for the given convex surface data, we can restrict the parameters so that each RCS ZFIF that we used to generate the surface RCZFIS is convex. Hence, using [29], we have the following theorem:

**Theorem 8.** *For given convex surface data* $\Gamma = \{(x_i, y_i, z_{i,j}, z_{i,j}^x, z_{i,j}^y) : i \in \mathbb{N}_N, \ j \in \mathbb{N}_M\}$, *if the shape parameter vectors and the scaling vectors satisfy* (26) *and* (27) *for each* $i \in \mathbb{N}_N$ *and* $j \in \mathbb{N}_M$, *then the corresponding bicubic partially blended RCZFIS* $\mathcal{F}_{\epsilon}^{\alpha}$ *will be convex on* $R = I \times J$.

**Numerical algorithm** for generating a convex RCZFIS for given convex surface data $\Gamma = \{(x_i, y_i, z_{i,j}, z_{i,j}^x, z_{i,j}^y) : i \in \mathbb{N}_N, \ j \in \mathbb{N}_M\}$:

- Step 1: Split $\Gamma$ into convex univariate Hermite datasets along the $x$-axis and $y$-axis.
- Step 2: Fix the values of signature vectors.
- Step 3: Choose the shape parameters and scaling factors as restricted in (26) and (27).
- Step 4: Construct the convexity-preserving univariate RCS ZFIFs along these univariate datasets with the parameters chosen in Step 2 and Step 3.
- Step 5: Construct a RCZFIS using these convex RCS ZFIFs and cubic blending functions $\Omega_0$ and $\Omega_1$ in (24).

The RCZFIS constructed in Step 5 is a convex surface interpolating the given convex data $\Gamma$.

**Remark 2.** *One can construct concave RCZFISs for given concave surface data with a similar procedure.*

**Example 4.** *We generated convex Hermite surface data on* $\{-1, \frac{-1}{3}, \frac{1}{3}, 1\} \times \{-1, \frac{-1}{3}, \frac{1}{3}, 1\}$ *from a bivariate function* $Z_f(x, y) = x^2 + y^2$. *We plotted Figure 5a,b using the zipper IFSs parameters given in Table 6. For Figure 5a, we chose random shape parameters and scaling functions, and the*

*corresponding RCZFIS is not convex on* $[-1, 1] \times [-1, 1]$, *which we can observe from its xz-view and yz-view plotted in Figure 5c,e, respectively. However, when we restrict the shape parameters and scaling function as prescribed in Theorem 8, the corresponding RCZFIS plotted in Figure 5b becomes convex on* $[-1, 1] \times [-1, 1]$, *which one can examine from its xz-view and yz-view plotted in Figure 5d,f, respectively.*

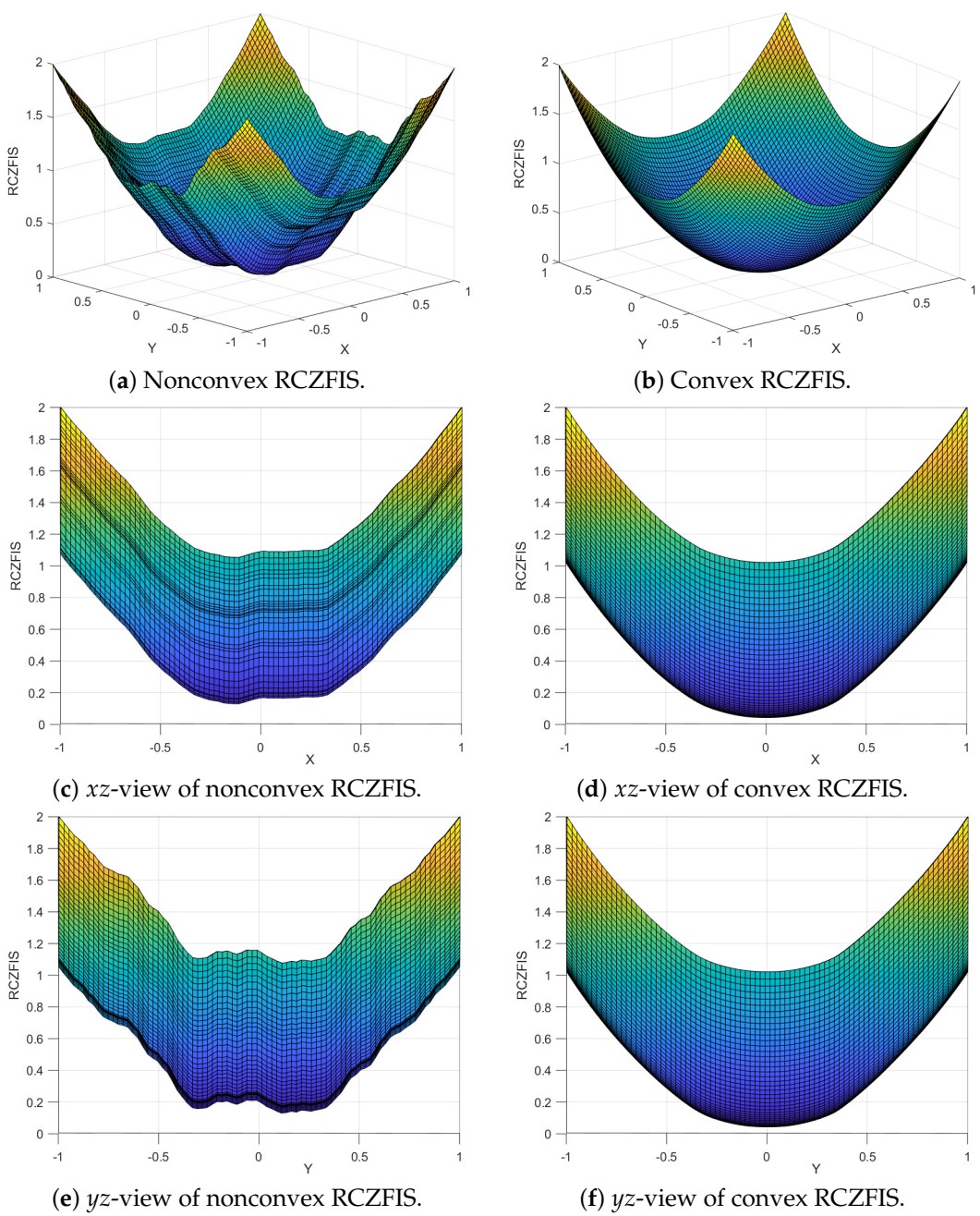

(**a**) Nonconvex RCZFIS.

(**b**) Convex RCZFIS.

(**c**) *xz*-view of nonconvex RCZFIS.

(**d**) *xz*-view of convex RCZFIS.

(**e**) *yz*-view of nonconvex RCZFIS.

(**f**) *yz*-view of convex RCZFIS.

**Figure 5.** Verification of convexity by RCZFIS.

**Table 6.** Zipper IFSs parameters for Figure 5.

| Figure 5a | Figure 5b |
|---|---|
| $\sigma^1 = \sigma^2 = \sigma^3 = \sigma^4 = (0.1, 0.06, 0.3),$ | $\sigma^1 = \sigma^2 = \sigma^3 = \sigma^4 = (1, 2, 3),$ |
| $\eta^1 = \eta^2 = \eta^3 = \eta^4 = (0.3, 0.2, 0.6),$ | $\eta^1 = \eta^2 = \eta^3 = \eta^4 = (3, 2, 5),$ |
| $\epsilon^1 = \epsilon^2 = \epsilon^3 = \epsilon^4 = (1, 0, 0),$ | $\epsilon^1 = \epsilon^2 = \epsilon^3 = \epsilon^4 = (1, 0, 0),$ |
| $\alpha^1 = \alpha^2 = \alpha^3 = \alpha^4 = (0.3, 0.25, 0.3),$ | $\alpha^1 = \alpha^2 = \alpha^3 = \alpha^4 = (0.02, 0.05, 0.02),$ |
| $\overline{\sigma}^1 = \overline{\sigma}^2 = \overline{\sigma}^3 = \overline{\sigma}^2 = (0.1, 0.06, 0.3),$ | $\overline{\sigma}^1 = \overline{\sigma}^2 = \overline{\sigma}^3 = \overline{\sigma}^2 = (1, 2, 3),$ |
| $\overline{\eta}^1 = \overline{\eta}^2 = \overline{\eta}^3 = \overline{\eta}^4 = (0.3, 0.2, 0.6),$ | $\overline{\eta}^1 = \overline{\eta}^2 = \overline{\eta}^3 = \overline{\eta}^4 = (3, 2, 5),$ |
| $\overline{\epsilon}^1 = \overline{\epsilon}^2 = \overline{\epsilon}^3 = \overline{\epsilon}^4 = (0, 1, 1),$ | $\overline{\epsilon}^1 = \overline{\epsilon}^2 = \overline{\epsilon}^3 = \overline{\epsilon}^4 = (0, 1, 1),$ |
| $\overline{\alpha}^1 = \overline{\alpha}^2 = \overline{\alpha}^3 = \overline{\alpha}^4 = (0.2, 0.25, 0.2),$ | $\overline{\alpha}^1 = \overline{\alpha}^2 = \overline{\alpha}^3 = \overline{\alpha}^4 = (0.02, 0.05, 0.05).$ |

## 7. Conclusions

This article introduces a class of novel continuously differentiable surface interpolants (RCZFISs) on a rectangular grid. The proposed surface interpolant is based on 16 parameters (8 shape parameters, 4 scaling factors, and 4 signature components) and two blending functions on each rectangular patch. It was observed that the RCZFIS converges the continuously differentiable data generated function uniformly. It can capture the irregularities associated with the partial derivatives of the data-generating function. For the fixed signature vectors, one can preserve the convexity of the data through RCZFIS using mild conditions on shape parameters and scaling functions. Some numerical examples have been given so the reader can become more familiar with these interpolants. The proposed scheme is more appropriate for modeling smooth convex surfaces with irregular partial derivatives in many areas of science and engineering. In automobile industries, convex and concave surfaces are used extensively for the shape of a vehicle. By varying signature only, one can design a wide variety of vehicles with very much similar shapes based on a variation on 1st order partial derivatives. The existence of minimum energy among $C^2$ zipper fractal splines, when the scale vector is fixed, is an open problem. The monotonicity- and convexity-preserving zipper fractal surfaces directly from zipper IFSs are open problems. The zipper fractal rational splines can be used as solutions of differential and integral equations if the associated problem contains a continuous and nowhere differentiable function.

**Author Contributions:** Conceptualization, V. and A.K.B.C.; methodology, V. and A.K.B.C.; software, V.; validation, V. and A.K.B.C.; formal analysis, V.; writing—original draft preparation, V.; writing—review and editing, A.K.B.C.; supervision, A.K.B.C. All authors have read and agreed to the published version of the manuscript.

**Funding:** This research received no external funding.

**Data Availability Statement:** The manuscript does not contain any data from a third party. All of the material is owned by the authors and/or no permissions are required.

**Conflicts of Interest:** The authors declare no conflict of interest.

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
