# Peer review of "Convexity-Preserving Rational Cubic Zipper Fractal Interpolation Curves and Surfaces"

_mca, doi:10.3390/mca28030074_

Round 1
Reviewer 1 Report
The authors propose a new and very effective method of construction of continuously differentiable surface interpolants (RCZFISs) on a rectangular grid, which are based on 16 parameters (8 shape parameters, 4 scaling factors, and 4 signature components) and two blending functions on each rectangular patch. This RCZFIS converges the continuously differentiable data generated function uniformly. The paper is carefully written and should be recommended for publication in Mathematics
Author Response
The authors are grateful to the reviewers for their valuable time, their utmost care about our manuscript, and for finding it pertinent.
Reviewer 2 Report
Some novel classes of continuously differentiable convexity-preserving zipper fractal interpolation curves and surfaces are introduced. Rigorously, portions of the technique are not detailed. There is no need to introduce the `vertical' scaling factor in l. 6. Curves are not functions as stated in l. 25. What does it mean "...and its fractal" in l. 52? Hermite interpolation data must be explained. What are the d_i and why can they calculated by the data? Why zipper? What was the motive to name them so? The a-fractal functions must be explained. the attractor or fractal is wrong in l. 98. What is the convexity feature of the data? Better use `univariable', `bivariable' functions, but `univariate' lines, 'bivariate' surfaces. Why bicubic partially blended?
Reviewer 3 Report
In the related manuscript, a class of novel continuously differentiable surface interpolants (RCZFISs) on a rectangular grid is introduced. The aimed surface interpolant has based on 16 parameters (8 shape parameters, 4 scaling factors, and 4 signature components) and two blending functions on each rectangular patch. It has been observed that the RCZFIS converges the continuously differentiable data generated function uniformly. It can capture the irregularities associated with the partial derivatives of the data-generated function. For the fixed signature vectors, one can preserve the convexity of the data through RCZFIS using mild conditions on shape parameters and scaling functions. Some numerical examples have been given so the reader can get more familiar with these interpolants. The proposed scheme is more appropriate for modeling smooth convex surfaces with irregular partial derivatives in many areas of science and engineering.
The quality of the manuscript is well.
The results obtained in this manuscript will be useful for the next studies.
All results derived in this paper are correct and the language of the paper is acceptable.
So, the manuscript deserves publication in this valuable journal after the authors check this manuscript word by word for grammar errors and punctuation.
Author Response
The authors are grateful to the reviewers for their valuable time, their utmost care about our manuscript, and for finding it pertinent.
We have checked the grammar, typos, and corrected several English sentences throughout the paper. We believe that it is error-free at this stage.
Reviewer 4 Report
Please see the attached file for the detailed comments.
